# CHRONOBERG: CAPTURING LANGUAGE EVOLUTION AND TEMPORAL AWARENESS IN FOUNDATION MODELS

## ABSTRACT

Large language models (LLMs) excel at operating at scale by leveraging social media and various data crawled from the web. Whereas existing corpora are diverse, their frequent lack of long-term temporal structure may however limit an LLM's ability to contextualize semantic and normative evolution of language and to capture diachronic variation. To support analysis and training for the latter, we introduce Chronoberg, a temporally structured corpus of English book texts spanning 250 years, curated from Project Gutenberg and enriched with a variety of temporal annotations. First, the edited nature of books enables us to quantify lexical semantic change through time-sensitive Valence-Arousal-Dominance (VAD) analysis and to construct historically calibrated affective lexicons to support temporally grounded interpretation. With the lexicons at hand, we demonstrate a need for modern LLM-based tools to better situate their detection of discriminatory language and contextualization of sentiment across various time-periods. In fact, we show how language models trained sequentially on Chronoberg struggle to encode diachronic shifts in meaning, emphasizing the need for temporally aware training and evaluation pipelines, and positioning Chronoberg as a scalable resource for the study of linguistic change and temporal generalization. Disclaimer: This paper includes language and display of samples that could be offensive to readers. Open Access: Chronoberg will be available publicly on HuggingFace.

## 1 INTRODUCTION

Language evolves continuously, reflecting shifts in knowledge, culture, and social norms. However, most large language models (LLMs) are trained on near-stationary datasets. Although highly effective at enabling LLMs at large-scale (Raffel et al., 2020; Lu et al., 2024), existing diverse web-crawled corpora, such as Common Crawl and Wikipedia, at best feature short-horizon temporal variations. Alas, without temporal grounding, language models risk conflating historical and contemporary meanings, for example, misinterpreting phrases such as "*Where is the woman to strew the flowers?*" by applying modern connotations. Such misreadings can distort semantic understanding, but may also amplify outdated stereotypes and ethical blindspots (Blodgett et al., 2020). This challenge is evident in hate speech detection models (Liu et al., 2019; Lees et al., 2022), where contemporary classifiers often fail to identify discriminatory language in historical contexts. As language and societal norms continue to evolve, it becomes increasingly critical to understand how models adapt their representations to ongoing and possible future linguistic change (Dhingra et al., 2022). Addressing both retrospective and prospective concept drift is key to the responsible development of temporally robust AI systems.

For such temporal contextualization, collections of edited books as a form of curated archive provide a more suitable resource. In fact, (Michel et al., 2011) has previously analyzed a large host of interesting "culturomics" patterns emerging from books, including several insights on the changes in grammar, lexicography, and the historical evolution of the collective mind of a culture. Unfortunately, obtained insights cannot be trivially captured in modern LLMs, as the underlying representations are not directly amenable to machine learning. Public representations like n-grams discard valuable sentence and paragraph context, whereas raw text archives like Google Books, Early English Books Online (EEBO) (ProQuest, 2008), and Project Gutenberg (Project Gutenberg) often lack the structured annotations required to study semantic drift and cultural shifts (Hamilton et al., 2016; Kutuzov et al., 2018) in temporal adaptation at scale.

In order to enable both the analysis and training of LLMs on a long-term timespan at scale, we introduce Chronoberg. Chronoberg is a diachronic dataset containing 2.7B (billion) tokens and spanning over 250 years of full-length English literary texts originating from Project Gutenberg (Project Gutenberg), which we have annotated temporally. A core contribution is the construction of temporally calibrated Valence-Arousal-Dominance (VAD) lexicons, which enrich the raw corpus with structured semantic and diachronic metadata. To this end, we extend the static NRC VAD lexicon (Mohammad, 2018; 2025) to include nearly 300,000 words across time. These scores allow for coarse sentiment tracking and support diachronic analysis of affective meaning, while also providing a structured benchmark for evaluating the temporal robustness of LLMs. We complement these lexicons with sentence-level annotations of sentiment trends based on the VAD scores, as well as outputs from modern LLM-based hate-speech detectors (Liu et al., 2019; Lees et al., 2022). In turn, the creation of Chronoberg empowers us to analyze how hate-speech detectors conflate modern connotations with historical reality and how contemporary language models perform under temporal shift. To this end, we train LLMs sequentially over several time periods and as a key result, expose that they struggle significantly with forgetting of prior information and generalization to future sentences that include terms which our historically calibrated VAD lexicons have identified to be particularly volatile.

## 2 RELATED WORK: LEXICAL SEMANTIC ANALYSIS AND TEMPORAL DATA

Diachronic linguistic datasets, such as Early English Books Online (EEBO) (ProQuest, 2008; Partnership, 2008) and the Google Ngram Corpus, pioneered large-scale cultural and lexical analyses. The employed n-gram–level formats have previously enabled seminal studies (Michel et al., 2011), laying out the groundwork for analyzing quantitative phenomena at the interface of social sciences and humanities. However, the lack of sentence context and semantic annotations at an n-gram level limits the datasets' utility for more timely LLM model training and evaluation. Other diachronic corpora, such as COCA (Davies, 2015), COHA (Mark, 2012), and CCOHA (Alatrash et al., 2020), are valuable for capturing American English variation, but remain fairly small in scale. Newer resources like TiC-LM (Li et al., 2025) and TemporalWiki (Jang et al., 2022) emphasize contemporary factual content rather than long-term semantic analysis. Chronoberg complements these prior efforts by providing full-length texts with temporal metadata, facilitating both semantic and affective analysis at yearly granularity in the context of modern-day LLMs.

Respectively, methodologies for studying semantic change have shifted from qualitative and manual linguistic analysis (Michel, 1897; Ullmann, 1962) to more quantitative, large-scale distributional approaches. Seminal methods include positive point-wise mutual information (Bullinaria & Levy, 2007), singular value decomposition (SVD) (Levy et al., 2015), and Word2vec (Mikolov et al., 2013). Alignment techniques like Compass-aligned distributional embeddings (CADE) (Di Carlo et al., 2019) have enabled temporal comparisons. These techniques are further supported by influential factor analysis (Osgood et al., 1957; Russell, 2003), which has led to the creation of VAD lexicons (valence: positive/negative word nature - arousal: active/passive tone - dominance: dominant/submissive word nature) with human-annotated scores for 45.000 contemporary English words (Mohammad, 2018; 2025). However, these lexicons are synchronic in nature; the scores reflect only contemporary linguistic understanding and do not account for historical semantic evolution. Consequently, the resource cannot track the changing connotations of words such as broadcast or febrile, as modern ratings fail to capture their historical usage (Perc, 2012). Chronoberg leverages VAD dimensions and contributes computationally constructed temporally aligned VAD lexicons, which are then used for sentence-level VAD annotations of the entire dataset to support the study of affective change and temporal robustness.

The increased need for such resources has been pointed out by select works, emphasizing how the changing nature of sentiments towards social groups is embedded in AI systems (Mendelsohn et al., 2020; Queerinai et al., 2023), while works focused on moderation drift and youth slang (Keidar et al., 2022; Mehta & Giunchiglia, 2025) highlight how rapidly changing language can undermine model robustness. Indeed, the emerging fields of continual learning (CL) (McCloskey & Cohen, 1989; Thrun, 1998; Mundt et al., 2023; Wang et al., 2024) and machine unlearning (Cao & Yang, 2015; Geng et al., 2025) are respectively concerned with training and evaluating models' ability to encode or deliberately remove knowledge over time. However, existing benchmarks (e.g., TOFU (Maini et al., 2024), WMDP (Li et al., 2024), MUSE (Shi et al., 2025), CL-Gym (Mirzadeh & Ghasemzadeh, 2021), CLEAR (Lin et al., 2021)) are either purely synthetic, small-scale, or lack temporal depth.

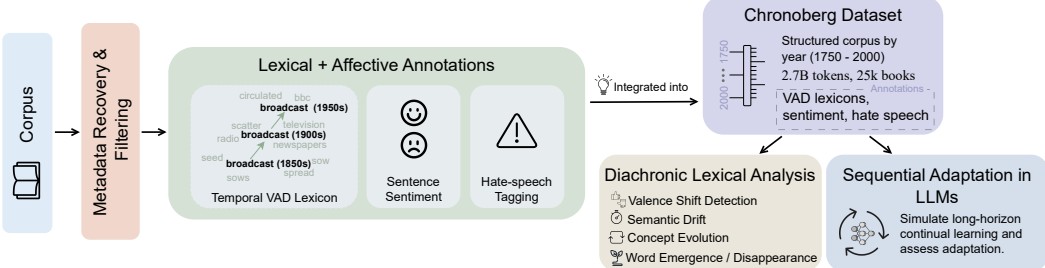

Figure 1: Overview of the Chronoberg dataset pipeline, spanning corpus curation, metadata recovery, and diachronic lexical analysis. The resulting annotations, including VAD lexicons and sentiment scores, form an integral part of the dataset and support downstream machine learning investigation.

Recent position papers (Verwimp et al., 2024; Mitchell et al., 2025) have thus called out for more realistic benchmarks. By providing a large, temporally annotated corpus with affective annotations in the form of VAD lexicons, Chronoberg introduces such a more natural application to benchmark CL, unlearning, and general temporally-adaptive machine learning strategies.

## 3 CHRONOBERG DATASET

In this section, we describe the pipeline developed to construct Chronoberg, encompassing the data selection, metadata inference, filtering stages, and VAD lexicon construction. These steps are depicted in Figure 1 and detailed in the following subsection. We note that each component is designed to maintain temporal consistency and ensure interoperability with contemporary NLP tools.

### 3.1 COLLECTION METHODOLOGY

The search for a large, openly accessible corpus of literary texts amenable to LLM training and evaluation leads us to (Project Gutenberg), which provides copyright-free English works in plain text and HTML formats, along with extensive metadata accessible via an API. However, original publication dates are frequently absent or inaccurate, often reflecting the digitization process rather than the first edition. The latter issue poses a significant challenge for temporal analysis.

**Metadata Recovery.** We developed an inference pipeline to assess publication dates by combining internal metadata with queries to external bibliographic sources (such as OpenLibrary and Wikipedia). Inferred publication years were further consistency checked to lie within the identified author's lifespan.

To verify its utility, we manually checked publication year estimates for 100 randomly sampled books spanning 1611–1912. OpenLibrary provided the best overall performance, with a mean absolute error (MAE) of $\pm 3.05$ years and standard deviation (SD) of $5.20$ years, the latter reflecting the disproportionate influence of a small number of outliers. Other sources, like Wikipedia, underperformed, providing lower coverage and significantly higher error rates. While majority voting yielded slightly better recall-based metrics, it resulted in a higher MAE (4.05 years) and lacked scalability due to inconsistent overlap among predictors. Google Books was excluded from large-scale inference due to restricted API access. Although even the best performing predictor yields an uncertainty of 3-5 years, we argue that the error margin is acceptable, as our diachronic analyses operate at the scale of decades rather than years, comfortably exceeding the typical variance introduced and would not alter the direction or conclusion drawn from diachronic trends. Nevertheless, this noise is inevitable as a practical design constraint of publication-date inference, and therefore we strongly recommend that future studies adopt appropriately coarse temporal bins no smaller than 15 years, which provides sufficient tolerance to predictor error and ensures robustness even under worst-case year misassignment. Full details for publication date inference are in Appendix A.2.

**Filtering.** Building on the validation results presented above, we adopted OpenLibrary as the default inference source, owing to its balance of coverage, recall, and scalability. Books without an inferred publication year or a known author were excluded, and inferred author lifespan data were used to

discard works likely published posthumously (and thus potentially skewing our later analysis). To maintain linguistic coherence and adequate temporal coverage, we retained only English-language books published between 1750 and 2000, a period aligned with Late Modern English and sufficiently broad to support historical analysis. Following these steps, the chronological backbone of Chronoberg contains 25,061 out of 73,500 books in Project Gutenberg, allowing year-by-year aggregation, annotation, and subsequent analysis.

### 3.2 LEXICAL AND AFFECTIVE ANNOTATIONS IN CHRONOBERG

In order to analyze diachronic shifts, we create lexical annotations and use them to construct a novel set of temporally aligned lexicons spanning Valence, Arousal, and Dominance (VAD) dimensions across time. Both the word-level lexicons for approximately 300,000 words and sentence-level VAD scores for the full corpus are released as part of Chronoberg.

**Temporal VAD Lexicons.** Building on prior linguistic studies investigating semantic change (Hamilton et al., 2016), our methodology uses diachronic distributional semantics to model shifts in word meaning. The core principle is to learn a high-dimensional vector for each word from its co-occurrence patterns within a given time period. To achieve this, we first train separate Word2Vec (Mikolov et al., 2013) models on a temporal slice of the Chronoberg corpus. Following validation to select suitable hyper-parameters, training for each interval has been conducted for 10 epochs with a context window size of 5 tokens on either side and an embedding vector dimensionality of 300. As noted before, each of these temporal slices corresponds to a 50-year interval between 1750 and 2000, with intervals chosen according to the observed variance in publication date estimation and including a reasonable safety buffer. We note that the boundaries do not correspond to any specific historical eras or expert-defined periodization and leave these to future work. Subsequently, these distinct embedding spaces are aligned using Compass Aligned Distributional Embeddings (CADE) (Di Carlo et al., 2019), which facilitates the direct comparison of word vectors across different decades.

We chose Word2Vec + CADE for being computationally efficient over 250 years of data and reliable for diachronic nearest-neighbour retrieval. This aligns with our VAD propagation method, which requires stable vector-space alignment to preserve local geometric consistency for cross-temporal semantic neighbourhood retrieval. While transformer embeddings (Devlin et al., 2019) could offer richer context, we do not expect fundamental shifts in VAD interpretation, as the core compression principles remain unchanged, and therefore leave this as future work.

We then estimate the VAD score for each target $(w)$ by selecting the Top-K nearest neighbors $(\mathcal{N}_k(w))$ in the embedding space and averaging their corresponding VAD values from the human-annotated NRC VAD lexicon (Mohammad, 2018; 2025):

$$\mathcal{N}_k(w) = \text{Top}k_{u \in \mathcal{V}_{\text{VAD}} \setminus \{w\}} \ s(e_w, e_u), \qquad \widehat{\mathbf{A}}_{\text{VAD}}(w) = \frac{1}{|\mathcal{N}_k(w)|} \sum_{u \in \mathcal{N}_k(w)} \mathbf{A}_{\text{VAD}}(u). \tag{1}$$

Here, $\mathbf{e}_w \in \mathbb{R}^d$ is the embedding of target word $w$, $s(e_w, e_u)$ is the cosine similarity, and $\mathcal{V}_{\text{VAD}}$ the set of words with NRC VAD annotations $\mathbf{A}_{\text{VAD}}(u) \in \mathbb{R}^3$. We thus make use of the human-annotated scores in the NRC VAD lexicon, which primarily reflect contemporary interpretations of words, but re-contextualize them computationally to account for historical contexts or diachronic semantic shifts that certain words may have undergone. To this end, we use modern valence scores as anchors to detect relative semantic drift. While this may introduce systematic anachronisms, it is unavoidable, and presents a natural choice, as retrospective annotations from humans several hundreds of years ago are not possible and few, if any, historians can operate at such scale.

We thereby present temporally adjusted lexicons in an extension to 335,804 words, assigning a real-valued score between -1 and 1 for the three dimensions. A key challenge in this process lies in determining a suitable number of top-K neighbors. Selecting too few (top-10) can miss contextual diversity, but taking into account too many (top-500) may lead to semantic over-smoothing and can introduce noise. Following empirical analysis, averaging the scores from the top-20 neighbors seems to mitigate these adverse effects. To determine the number of neighbors required to retrieve 20 known words from the lexicon, we analyzed the retrieval rates for 100 high- and low-valence anchor words from Chronoberg. This analysis revealed that retrieving 20 known neighbors typically necessitates inspecting the top-100 nearest neighbors, see Appendix A.3 for a more detailed discussion.

Figure 2: Sentence-level VAD scoring in Chronoberg. Instead of simple averaging across all words, we introduce a two-stage aggregation strategy: (i) part-of-speech–based averaging over adjectives, verbs, and adverbs (JJ/VB/RB), followed by (ii) clause-based scoring, which computes clause-level valence means and selects the most extreme value. These complementary steps build on one another to form the final sentence-level scoring procedure in Chronoberg, enabling robust detection of affective polarity in complex contexts.

**Sentence-level VAD Annotations.** Extending our analysis beyond individual words, we leverage our five sets of temporal VAD lexicons to assess sentiment in larger blocks of text, especially sentences. However, a naive aggregation of word scores in a sentence can lead to a neutrality bias, given the high frequency of neutrally perceived words in language. To mitigate this issue, as illustrated in Figure 2, we introduce two modifications to our scoring pipeline: JJ/VB/RB and clause-based averaging.

*JJ/VB/RB averaging:* First, we focus only on emotionally salient parts of speech (in particular verbs (VB), adverbs (RB), and adjectives (JJ)), thereby minimizing the influence of neutral words on the final sentence score.

*Clause-based averaging:* Second, our method accounts for sentiment variations within a sentence, following prior work by (Wang et al., 2018). Accordingly, we calculate an average valence score for each clause, based on its adjectives (JJ), verbs (VB), and adverbs (RB). The overall sentence score is then assigned as the minimum value among these clause-level scores.

*Final Score:* We compute the final score $\widehat{\mathbf{I}}_{\text{VAD}}(\text{sent})$ by combining these two averaging approaches:

$$\widehat{\mathbf{A}}_{\text{VAD}}(\text{sent}) = \min_{C_i \in \text{Clauses}} \left( \frac{\sum_{t \in C_i,\ \text{pos}(t) \in \{\text{JJ,VB,RB}\}} \mathbf{A}_{\text{VAD}}(t)}{N_{C_i, \{\text{JJ,VB,RB}\}}} \right) \quad (2)$$

where $\mathbf{A}_{\text{VAD}}(t)$ represents the VAD scores and pos(t) is the part of speech tag for each token in the sentence. Since values range from -1 to +1, we will consider the sign to carry a respective connotation for simplicity in further analysis. We believe this to be justified by focusing on assessment of the overall change in consecutive analysis. However, we acknowledge that perceived connotation can depend on various subjective factors. Our sentence-level valence annotations are a core component of Chronoberg and are publicly available to support transparency and further analysis.

## 3.3 DATASET COMPOSITION & STATISTICS

Finally, we summarize Chronoberg's composition and highlight statistics underlying diachronic shifts. A dataset sheet for datasets (Gebru et al., 2021) is provided in Appendix D.

**Composition.** Overall, Chronoberg is composed of 2.7B tokens, representing 91M sentences from 25,061 English-language books published between 1750 and 2000, with additional metadata in the form of temporally-aligned VAD lexicons that span 335,804 words. On average, approximately 28% of sentences per 50-year epoch are classified as *negative* (valence $< 0 - \epsilon$), while 50% are *positive* (valence $> 0 + \epsilon$), considering $\epsilon$ to be 0.05. Notably, some unique samples across epochs exhibit a change in average valence scores, indicating affective drift.

**Statistics of Diachronic Shifts in Words & Sentences in Chronoberg.** Figure 3 reports the extent of valence shifts between the selected epochs. While the majority of words exhibit stable affective

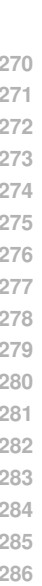

Figure 3: Distribution of valance shifts between consecutive 50-year intervals in Chronoberg. Bars show the count of words and sentences undergoing changes from positive to negative and vice versa, across different thresholds of rate of change. While most words remain effectively stable, thousands of samples exhibit substantial shifts in both directions.

Table 1: Examples of words with strong diachronic valence shifts in Chronoberg. The left column reports words that transitioned from positive to negative connotations, while the right column highlights their opposites. Scores are averaged over the top-20 nearest neighbors per epoch, illustrating how semantic and affective associations evolve across centuries, approximated by 50-year time intervals.

| | _Positive → Negative_ | | | | | | _Negative → Positive_ | | | | |
|---|---|---|---|---|---|---|---|---|---|---|---|
| **Words** | **1750s** | **1800s** | **1850s** | **1900s** | **1950s** | **Words** | **1750s** | **1800s** | **1850s** | **1900s** | **1950s** |
| asylum | 0.27 | -0.24 | -0.54 | -0.52 | -0.65 | febrile | -0.58 | -0.53 | -0.66 | -0.54 | 0.33 |
| germs | 0.15 | 0.26 | -0.14 | -0.55 | -0.61 | infatuation | -0.66 | -0.63 | -0.52 | -0.35 | 0.40 |
| homeless | 0.11 | -0.62 | -0.63 | -0.66 | -0.56 | destiny | -0.54 | 0.06 | 0.32 | 0.11 | 0.44 |
| punk | 0.20 | 0.14 | -0.25 | -0.17 | -0.26 | bravo | -0.37 | 0.34 | 0.42 | 0.53 | 0.60 |
| weird | 0.30 | 0.01 | -0.28 | -0.33 | -0.43 | bewitchments | -0.44 | -0.44 | -0.2 | -0.2 | 1.0 |

meanings over time, we identify 7,885 words that shifted from positive to negative and 8,787 words that shifted from negative to positive. Notably, the 7,000 most variable words alone contribute to contextual changes in more than 90,000 sentences within Chronoberg, underscoring both the richness and the analytical potential of the dataset.

Table 1 presents representative examples of words with the highest degree of change. Words such as _homeless_ and _germs_, which in the most recent epoch (1950s) carry negative valence scores (-0.61 and -0.56, respectively), historically expressed more neutral or even positive meanings. For example, _germs_ originally meant a seed or origin of life, but primarily now refers to pathogens. Similarly, _homeless_ shifted from a neutral term to being strongly associated with poverty. Conversely, _febrile_ (-0.58 → 0.33) moved from being a medical term to emotionally charged contexts. _Infatuation_ softened from delusion to convey romantic admiration, while _destiny_ shifted from threatening fate to purpose or self-realization. These examples demonstrate how embedding-based VAD scoring provides interpretable, fine-grained insights into the affective trajectory of words. They further emphasize the value of our temporal VAD lexicons and the Chronoberg dataset as instruments for analysing semantic and affective change over time. Additional examples of words exhibiting diachronic valence shifts are provided in Appendix A.4.

## 4 EXPERIMENTS AND DATASET ANALYSIS

To showcase Chronoberg's utility, we demonstrate experiments that highlight the introduced VAD lexicons' role in tracing sentence-level semantic shifts in connotations across different time periods, examine whether modern hate check tools handle these shifts, and finally, investigate how sequentially trained language models adapt to the evolving concept drifts. In the following subsections, we organize the experiments into coarse 50-year intervals, which were chosen for simplicity and to

Table 2: Sentence-level classifications in Chronoberg using LLM-based hate-check tools (RoBERTa+Perspective API, OpenAI Moderation API) (🚩 = Hate, ✓ = Non-hate) and valence-based scoring (👎 = Negative, 👍 = Positive sentiment). Table 2a shows a comparison of sentence-level classifications in Chronoberg. Table 2b illustrate the extent to which modern LLM hate-check classifiers rely on surface-level lexical cues.

| YEAR | Sentences | Hate-Check Models RoBERTA+Persp | OpenAI | Valence Score | Affective Sentiment |
|------|-----------|:---:|:---:|:---:|:---:|
| 1750s | but i loathe you, you apache indian! | 🚩 | 🚩 | -0.50 | 👎 |
| | Where is the woman to strew the flowers? | 🚩 | ✓ | 0.14 | 👍 |
| | you horse-hair hypocrite, you! | ✓ | ✓ | -0.60 | 👎 |
| 1800s | How I wish that you were black!–I detest your colour. | 🚩 | 🚩 | -0.79 | 👎 |
| | The black parts of them must be cut away, when the skin is taken off. | 🚩 | ✓ | 0.02 | 👍 |
| | he redoubled his gayety and carelessness. | ✓ | ✓ | -0.71 | 👎 |
| 1850s | the irish are mostly very filthy and diseased. | 🚩 | 🚩 | -0.66 | 👎 |
| | They carry faggots | 🚩 | 🚩 | 0.08 | 👍 |
| | An outlandish pair brought from new york for some tory hussy | ✓ | ✓ | -0.10 | 👎 |
| 1900s | As a race, as a family, the blacks have no loyalty. | 🚩 | 🚩 | -0.02 | 👎 |
| | I may cut you out of my gold expedition, if you get gay. | 🚩 | 🚩 | 0.175 | 👍 |
| | We know how heartless he is, how vindictive, how horribly cruel. | ✓ | ✓ | -0.77 | 👎 |
| 1950s | I hope with the Negroes, Indians, .. reduce Virginia.. | 🚩 | 🚩 | -0.18 | 👎 |
| | Black should never be worn at a wedding. | 🚩 | 🚩 | 0.10 | 👍 |
| | Why are the Africans in Algeria rising against their white French oppressors? | ✓ | ✓ | -0.05 | 👎 |

(a) For each 50-year interval, we organize based on model agreement. The first row illustrates cases where all tools collectively classify an instance as harmful. Rows 2-3 show instances where they disagree, classifying them as either positive or negative. While VAD lexicons provide interpretable complementary signals, we acknowledge that harmful texts are inherently subjective; therefore, we do not regard them as definitive solutions to LLM misclassification, but rather as potential tools to enhance LLM performance.

| Sentences | Hate-Check Models RoBERTA+Persp | OpenAI | Valence Score | Affective Sentiment |
|-----------|:---:|:---:|:---:|:---:|
| In my way home to my tent, I saw a faggot lying in the way | 🚩 | 🚩 | 0.05 | 👍 |
| In my way home to my tent, I saw a ~~faggot~~ **firewood** lying in the way | ✓ | ✓ | 0.10 | 👍 |
| In my way home to my tent, I saw a ~~faggot~~ **sticks** lying in the way | ✓ | ✓ | 0.02 | 👍 |
| In my way home to my tent, I saw a ~~faggot~~ **bundle** lying in the way | ✓ | ✓ | 0.09 | 👍 |
| In my way home to my tent, I saw a ~~faggot~~ **turves** lying in the way | ✓ | ✓ | 0.08 | 👍 |

(b) We examine the shift in sentence-level classification by replacing a target word with synonyms from Word2Vec embeddings. Here, substituting 'faggot' with the appropriate 'firewood' in a 1850s sentence leads to more accurate classification, emphasizing the extent to which contemporary hate-speech detection systems remain sensitive to surface-level lexical cues. Quantitative results are reported in Table 3.

mitigate variance introduced by uncertainty in publication year estimates following prior descriptions. None of the evaluations with LLMs were provided with year of the text as context, reflecting real-world scenarios. Moreover, the OpenAI moderation API does not allow providing any additional context during inference.

Beyond annotating general affective drifts, we leverage our temporal VAD annotations to contextualize practical notions of harmful language over time. To this end, we evaluate contemporary hate-check tools on sentences across different time intervals and validate the sentiment using our lexicons. We choose hate speech since it represents a particularly well-defined subset that is inherently negatively connoted. This enables us to investigate the alignment between hate-check outputs and VAD scores, highlighting cases where our annotations accurately capture meaningful affective polarity. More importantly, discrepancies in sentiment expose where modern classifiers fail to recognize historically situated expressions of hate or over-generalize from present-day keyword associations.

To select meaningful baselines, we have started by considering nine different hate speech detection tools, as well as the seven most popular Hugging Face models at the time of writing. Using HateCheck (Röttger et al., 2021) (a suite of functional tests across several dimensions), we found that many approaches performed no better than chance. Notable exceptions were RoBERTa (Liu et al., 2019)

with the highest recall and the Perspective API (Lees et al., 2022) with the highest precision. We refer the reader to Appendix 2 for the full quantitative study. To combine tools' strengths in application to Chronoberg, we thus consider a two-stage pipeline as a meaningful modern hate-checker: RoBERTa first flags a broad set of potentially hateful sentences, which are then filtered by the Perspective API to reduce false positive counts. In addition, we have also employed the latest OpenAI Moderation API[1] as the recently popular contender.

In Table 2a we first show representative examples, where VAD annotations and current hate-check tools agree and diverge. They are illustrative of notable trends in how modern hate-check tools flag negative sentences. While the latter achieve consistent correct predictions in cases of explicit sentiment (first row of each time interval), they seem to struggle when sentiment is implied rather than directly stated. For instance, a neutral phrase such as *Black should never be worn at a wedding* is incorrectly flagged as hateful by both hate-check tools, whereas the valence scores more accurately capture its neutral sentiment (0.10). Another illustrative case from the 1850s is the phrase *tory hussy*, where LLMs misinterpret *hussy* with its modern connotation, which influences their judgment of the sentence's hatefulness. Our examples also corroborate prior literature's findings, for instance the well-known meaning and connotation shifts in the words *faggot* or *gay* (Michel et al., 2011).

Table 3: Analysis of changes in disagreement score between VAD lexicons and hate-speech classifiers. We observe a ∼59% disagreement b/w VAD and RoBERTa+Perspective, whereas ∼38% with the OpenAI moderation tool across different eras. Whereas substituting the target words into their modern synonyms, as in Table 2b, reduces disagreement by ∼28% for OpenAI Moderation API and ∼24% for RoBERTa+Perspective API, showing reliance of classifiers on surface-level lexical cues.

| Years | RoBERTa+Persp w/o substitution | OpenAI w/o substitution | RoBERTa+Persp w/ substitution | OpenAI w/ substitution |
|---|---|---|---|---|
| 1750–1799 | 57.9% | 38.2% | 44.0% (↓ 24.0%) | 27.1% (↓ 28.8%) |
| 1800–1849 | 59.9% | 39.4% | 45.4% (↓ 24.2%) | 27.3% (↓ 30.1%) |
| 1850–1899 | 60.6% | 38.7% | 46.6% (↓ 23.1%) | 27.9% (↓ 27.9%) |
| 1900–1949 | 59.4% | 37.8% | 44.8% (↓ 24.5%) | 27.3% (↓ 27.8%) |
| 1950–2000 | 50.6% | 34.2% | 38.2% (↓ 24.5%) | 24.0% (↓ 29.8%) |

With respect to discrepancies between hate-check tools and our temporal VAD lexicons, we hypothesize that current LLMs may rely too heavily on modern surface-level keywords. To gauge the scale of the latter effect, we quantify *disagreement* across the entirety of Chronoberg, which we define as the percentage of instances where predictions from the RoBERTa+Perspective API, OpenAI Moderation API and our VAD lexicons disagree, providing a way to assess how modern classifiers handle historical language. In Table 3, first we report raw disagreement rates. As we do not wish to define what "hate" is in absolute terms due to its subjective and complex nature, we extract all examples from Chronoberg that are considered hateful by the RoBERTa hate-checker. We find a ∼ 59% contradiction rate in the initial eras of 1750-1850s, as extracted hate cannot be positive. However, as we progress to the later eras, the rate of disagreement starts to reduce to 50%, suggesting hate-checkers are not only imperfect, but also struggle more with capturing meaning at a specific point in time. On the other hand, we find disagreement of ∼ 85% between the OpenAI Moderation API and RoBERTa, also highlighting general volatility of LLM tools.

Additionally, to illustrate the reliance of modern LLM hate-speech classifiers on surface-level keywords, we include a quantitative experiment (shown in Table2b that analyses how replacing words (e.g., "f*ggot" with an innocuous synonym such as "firewood" from Word2Vec) affects classifier outputs. We observe that replacing surface-level keyword with historically accurate, non-toxic neighbours meaningfully restores classifier predictions back to non-hate. Our findings showcase that Chronoberg's VAD lexicons, while not a moderation tool, provide a useful complementary signal for checking LLM predictions for changing contexts over time.

### 4.1 SEQUENTIAL ADAPTATION IN LLMS TRAINED ON CHRONOBERG

In complement to our earlier analysis, we now showcase Chronoberg's utility in investigating how well LLMs trained under different temporal regimes can adapt to semantic change. Specifically, we

---

[1]https://platform.openai.com/docs/guides/moderation

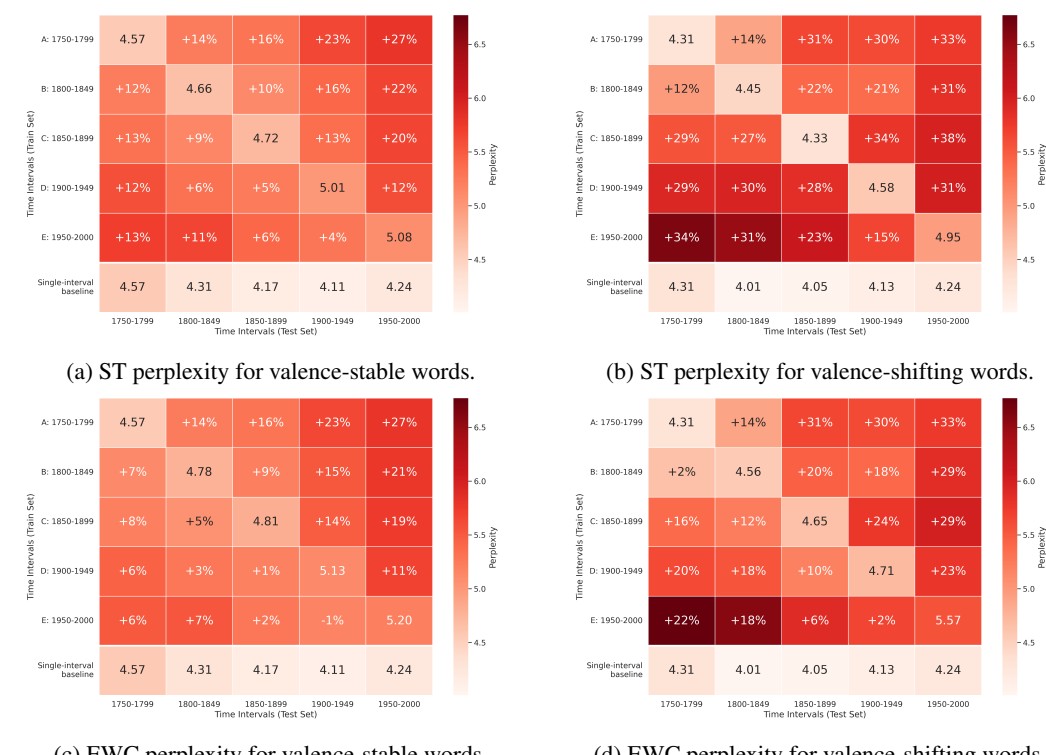

(a) ST perplexity for valence-stable words.

(b) ST perplexity for valence-shifting words.

(c) EWC perplexity for valence-stable words.

(d) EWC perplexity for valence-shifting words.

Figure 4: Perplexity of sequentially trained (ST) models and models trained continually with EWC, evaluated on test sets with words that have stable valence (a, c) and exhibit valence shift (b, d). Higher perplexity indicates worse language modelling performance. We observe that sequentially trained models suffer from forgetting (lower-left off-diagonal) significantly more on valence-stable words than valence-shifting ones. For instance, the model shows only a +13% rise on stable vs. +34% on shifting words at the end of sequential training for the initial content (row E: 1950-2000 for column A: 1750-1799). Similarly, generalization to new time intervals (upper-right off-diagonal) is significantly worse, especially in later time intervals. EWC is able to reduce catastrophic forgetting significantly (e.g. only a +6% rise vs. +13% with ST for the initial interval at the end). However, the reduction is much more prominent on valence-stable words than valence-shifting ones, where it remains hard to consolidate knowledge and to generalize. Similar results for LoRA are in Appendix Figure 10.

investigate whether LLMs reflect historical concept drift and temporal generalization. To this end, we trained 1.4B-parameter models from scratch using the Pythia architecture (Biderman et al., 2023). We trained models using NVIDIA A100-80GB GPUs under three distinct temporal setups designed to simulate long-horizon continual learning and assess strategies for adapting to future shifts in language and societal norms: **(1)** sequential training, where the model is trained incrementally on 50-year intervals of Chronoberg, **(2)** bin-based training, with separate models trained on individual 50-year bins to examine temporally localized learning, and **(3)** two continual learning baselines, namely the prevalent Elastic Weight Consolidation (EWC) (Kirkpatrick et al., 2017) and Low-Rank Adaptation (LoRA) (Hu et al., 2022). Detailed training configurations are provided in Appendix C.

To assess temporal robustness, we constructed two types of test sets for each 50-year interval based on diachronic valence trends: (a) sentences containing valence-stable words whose affective meaning remains constant over time, and (b) those exhibiting a clear valence shift. We assess model performance using *perplexity*, a standard measure of language model confidence that quantifies how well a model predicts the next word in a sequence. Lower perplexity indicates better fluency and alignment with expected language patterns. Ideally, a model would be able to learn from new experiences and maintain its knowledge of the past. However, our expectation is that a model maintains a baseline perplexity only for valence-stable words across time, with higher perplexity for valence-shifting words. Under naive sequential training, we expect the emergence of catastrophic forgetting (McCloskey &

Figure 5: Overview of a wide range of downstream applications enabled by Chronoberg, including diachronic lexical analysis, benchmarking sequential training of LLMs, and machine unlearning.

Cohen, 1989), leading to higher perplexity on earlier intervals after sequential training is complete. Here, we anticipate that valence-shifting words may be learned in the current interval, but generalize poorly across time. In contrast, CL methods are expected to better preserve knowledge, keeping perplexity low and consistent across intervals. We posit that present approaches will nevertheless yield higher perplexity for valence-shifting words in a struggle to capture semantic drift and consolidate inconsistent context.

We compare sequential training (ST) to EWC in Figure 4, but note that findings are consistent with LoRA as a continual learning method (see Appendix C). In panel 4a, we observe that sequential training yields perplexities that deteriorate mildly over time (bottom-left triangle values), whereas forward generalization in time (top-right triangle values) is more challenging when the temporal jump is large (e.g. from 1750 to 1950 with a 27% perplexity increase). However, this deterioration and lack of generalization is substantially exacerbated for valence-shifting words, as evident in panel 4a. In panel 4c, we confirm our earlier hypothesis that EWC (as a continual learning method) is indeed able to largely avoid forgetting, as observed perplexities on previously seen intervals remain much closer to the diagonal values (the performance on the current interval at the time) than for sequential training.

We can also see that forward generalization remains equally challenging, which is natural given that continual learning methods can only maintain the past. This degradation is asymmetric, since forward generalization deteriorates more sharply for valence-shifting words than for valence-stable ones. This is likely due to higher plasticity demands, that is, more optimization steps need to be taken to accommodate the semantic evolution across training epochs.

In panel 4d we again see that both temporal dimensions are exacerbated. Here, the continually learned model that is able to mitigate forgetting for valence stable content now also struggles significantly more with valence-shifts. The localized learning (diagonal) still performs well, but the nature of possibly temporally contradicting valence-shifts hinders even a continual learner from properly consolidating past knowledge, improving upon sequential training without fully resolving the issue. Our experiments thus position Chronoberg as an excellent resource to analyze realistic sequential learning strategies, highlighting their present insufficiency in capturing historical semantic drift and opening up development of novel techniques.

## 5 CONCLUSION

We introduced Chronoberg, a large-scale, temporally structured corpus of English books spanning the years 1750-2000, enriched with diachronic VAD lexicons and sentence-level affective annotations. Using these resources, we quantified shifts in affective meaning and demonstrated the need for modern LLM-based tools to better situate their detection of discriminatory language and contextualization of sentiment across various time-periods. Further, we showed that language models trained sequentially struggle to encode diachronic shifts in meaning, highlighting gaps that standard continual learning methods only partially address. Beyond the experiments presented here, Chronoberg opens several promising avenues. As illustrated in Figure 5, these include broader continual- and lifelong-learning studies with temporal training pipelines, as well as machine unlearning protocols to address historically contingent slurs or outdated facts. Future work should also explore decade-level or coarser temporal analyses, in particular in exploration of further interdisciplinary avenues that associate Chronoberg and tie its machine analysis to key historical eras or literary epochs.

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

APPENDIX

We have organized our supplementary material in the following order:

- **Section A: Dataset Curation and Lexical Analysis.** Additional details on the dataset collection, curation, and filtering pipeline, along with an extended analysis on VAD lexicons.
- **Section B: Extended Hate Speech Analysis.** Expanded evaluation of harmful texts in Chronoberg, including comparisons across multiple hate-speech detection tools, sentence-level analysis, and highlighting cases of disagreement between different models and valence scores in hate prediction.
- **Section C: Model Training and Experimental Setup.** Detailed description of the sequential training process of the large language models (LLMs) supplemented with implementation details, hyperparameters, and additional experimental results.

## A  ADDITIONAL CHRONOBERG DATASET DETAILS

This appendix complements main body Section 3 by providing additional information on the choice of data sources, available metadata, and recovery of publication dates through inference methods. Each of the following subsections expands upon specific components of the dataset pipeline to clarify design decisions and practical challenges. We also provide insights into the distribution and the thematic composition of Chronoberg.

### A.1  DATA SOURCE

As outlined in Section 3 of our dataset construction pipeline, we now present a detailed explanation of the rationale behind our choice of data corpus. The initial step in constructing a text dataset involves identifying an appropriate data source. For temporal datasets, two primary criteria are essential: first, the availability of timestamps indicating when the data was created; second, a substantial volume of content to ensure comprehensive coverage across different time periods. While large datasets are generally beneficial for language modelling since their performance improves with the amount of training data, temporal datasets require extensive data to capture variations over time.

In order to meet these requirements, we seek extensive text collections produced over the past centuries, easily accessible for research purposes, and accompanied by metadata detailing date of creation. Books emerged as a natural solution since they offer coherent and curated content, especially when compared to shorter form content like news or social media posts.

However, using books for large-scale, temporally annotated datasets presents several practical challenges:

1. Copyright Restrictions: Many books are under copyright restrictions, limiting free access to their full texts.

2. Digitization Requirements: To be usable, books must be available in digital formats.

3. Metadata Availability: Metadata, such as accurate author names and publication years, is crucial, since manually annotating hundreds of thousands of books without this information is unfeasible.

4. Programmatic Access: Efficient data collection necessitates programmatic interaction with the data source, such as through APIs, to download and filter relevant books in bulk.

We examined various online book databases, including Google Books (Michel et al., 2011), the Internet Archive, and Project Gutenberg (Project Gutenberg), to assess their suitability for large-scale historical text collection. Google Books offers an extensive online library, with full-text search across a vast collection of books and metadata such as publication dates. However, the API imposes significant constraints, limiting queries to a maximum of 40 results per request. Pagination requires numerous inefficient calls, and large-scale automated retrieval is hindered by protective measures such as CAPTCHAs. Moreover, the API requires a keyword-based search and does not permit queries based solely on publication year, making systematic dataset construction difficult.

The Internet Archive provides a large repository of digitized texts, often with richer metadata than Google Books. However, the quality and completeness of metadata varies considerably across the entries, and publication year information is frequently missing or unreliable. Additionally, bulk access is limited by rate restrictions and heterogeneity of formats, which further complicates large-scale preprocessing.

Project Gutenberg is a widely used digital library of public domain literary works, providing unrestricted access to full-length texts. Each entry is accompanied by metadata covering attributes such as *title, authors, subjects* and *issue date*, as shown in Table 4, together with other metadata fields available. We therefore select this as the primary resource for curating our Chronoberg dataset. However, key metadata, most notably publication dates, are frequently absent or inconsistent, which poses challenges for constructing a coherent, temporally stratified dataset. The methodological details for our curation process to address this are discussed in the following section.

Table 4: Project Gutenberg metadata fields with descriptions and catalogue availability. While many bibliographic attributes are present (✓), crucial information such as original publication dates is missing (✗), necessitating external inference for the construction of Chronoberg.

| Attribute | Explanation | In Catalogue |
|---|---|---|
| **ID** | A real number assigned by Project Gutenberg to uniquely identify the eBook | ✓ |
| **Type** | Text (>98 %), dataset, sound, image [...] | ✓ |
| **Issued** | Release date of the book | ✓ |
| **Title** | The title of the book | ✓ |
| **Language** | The language in which the book is available | ✓ |
| **Authors** | All authors of the eBook | ✓ |
| **Subjects** | Library of Congress subject headings | ✓ |
| **LoCC** | Library of Congress entries | ✓ |
| **Bookshelves** | Hand-curated eBook collections supplemented by 64 "Browsing" categories which were automatically assigned to mimic browsing in a bookstore | ✓ |
| **Publisher** | The publisher of the book | ✗ |
| **License & Rights** | Specifies the book's copyright status (e.g public domain in the USA) and, when applicable, the specific license governing its use. | ✗ |
| **Downloads** | How often the book has been downloaded | ✗ |
| **Birth/death dates** | Birth and death rates of all authors and translators if available | ✗ |
| **Description & MARC520** | Description and Summary of the eBook | ✗ |
| **Translators** | All translators of the eBook (if any) | ✗ |
| **Datatypes** | E.g. text, HTML, ePub, PDFs, [...] | ✗ |

## A.2 DATASET COLLECTION AND CURATION

Project Gutenberg contains some metadata inaccuracies, most notably the original year of publication often reflects the date of digitization rather than the actual release year. Accurate publication dates are critical for curating Chronoberg, as our aim is to order books chronologically for further study.

**Metadata and External Sources:** To address this limitation, we leverage available metadata attributes such as title, author, and the author's birth and death dates. We also queried multiple external sources such as Google Books (Google Books), Google Search, the Library of Congress API (Library of Congress Linked Data Service), Open Library (OpenLibrary), Wikipedia (Wikipedia), using book titles and author lifespans to determine the correct publication year. While dates explicitly mentioned in book titles can provide additional cues when available, this method is applicable only to a limited subset of works. To assess the reliability of these tools, we manually annotated 100 books from Project Gutenberg, evenly spread between 1611 and 1912. Due to access limitations, Google Books

Table 5: Comparison of publication year inference methods against 100 manually annotated ground truth samples. Accuracy is reported at different tolerances, with acc@0 representing the exact year, acc@5 within 5 years and acc@7 within 7 years. Open Library emerges as the most reliable predictor with broad coverage and low error (MAE = 3.05 years), making it the most reliable stand-alone source. Wikipedia achieves limited recall and lower accuracy. Majority voting across predictors offers marginal recall gains but does not scale well, reinforcing our choice of Open Library as the default predictor.

| Scores | Open Library | Wikipedia | Majority Vote |
|---|---|---|---|
| Values | 92 | 47 | 96 |
| Correct | 49 | 25 | 41 |
| Accuracy @0 (%) | 53.3 | 53.2 | 42.7 |
| Accuracy @5 (%) | 76.0 | 39.0 | 74.0 |
| Accuracy @7 (%) | 79.0 | 42.0 | 80.0 |
| MAE (years) | 3.05 | 3.36 | 4.05 |
| Standard Deviation (years) | 5.20 | 7.24 | 6.56 |

and Google Search were excluded, and the Library of Congress API was discarded because of poor performance.

Among the two remaining methods, shown in Table 5, Open Library was the most reliable, correctly estimating publication years for 49 out of 100 tested books, with a Mean Average Error (MAE) of 3.05 years and an accuracy (acc@5, i.e. within 5 years margin) of 76%, which falls within the acceptable tolerance range for our analysis. The Wikipedia API retrieved years for 25 books with an accuracy of 42.0%. However, due to the limited sample sizes for Wikipedia, its reliability remains uncertain. The Open Library API provided sufficiently accurate data to support the construction of a chronological text corpus, particularly because we constrained errors within the author's lifespan.

**Filtering:** The publication dates were constrained to fall within the author's year of birth and death as an additional consistency check. In case of missing information, a default lifespan of 100 years was assumed. When API methods returned multiple year estimates, the most frequently occurring year (i.e., the modal year) was selected; in the case of a tie, the first appearing year was chosen. Further filtration criteria were applied to refine Chronoberg:

(i) *Publication Year:* Books lacking any inferrable publication year (40.7%) were excluded. We restricted the dataset to books published between 1750 and 2000. This range balances token availability per decade and ensures linguistic consistency, as early modern English had largely been superseded by late modern English by 1750. Additionally, this timeframe allows for the analysis of historical shifts in public perception and hatespeech evolution by contrasting older texts with those from the modern era.

(ii) *Author Metadata:* Only books with a known author and recorded birth year were retained, leading to the exclusion of 22.9% of volumes. To ensure plausibility, we retained only works published within the author's lifespan; for authors with unknown death years, this corresponds to a default cut-off of 100 years after birth. This step removes posthumous editions and maintains consistency.

(iii) *Language:* The dataset was limited to English-language texts (80.3% of the total books to reduce cross-linguistic variation and maintain consistency in temporal language analysis.

(iv) *Translations:* Translated works (8.4%) were removed, since their publication dates often deviate substantially from the original text, potentially distorting historical trends.

(v) *Content Type:* Non-textual materials such as images and audio files (1.7%) were excluded to preserve a purely text-based corpus.

(vi) *Copyright and Availability:* Only books explicitly marked as public domain in the U.S. (98.8%) were included. Of these, 75 books lacked downloadable plaintext files in Project Gutenberg.

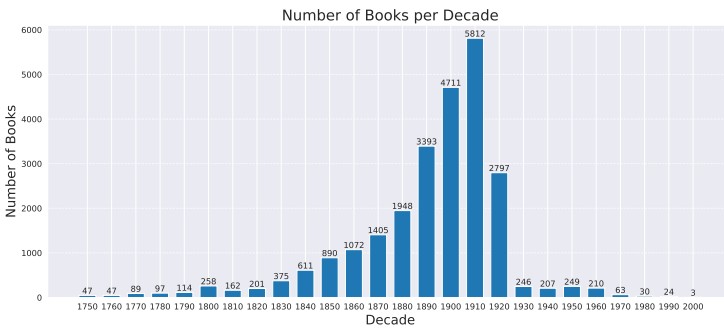

Figure 6: Temporal distribution of books in Chronoberg across decades (1750 - 2000). The number of available texts grows steadily until the 1920s, likely reflecting improved literacy and publication, but drops sharply thereafter due to copyright restrictions on works published after 1920. Consequently, early 20th century works dominate the distribution, while post-1920s coverage remains sparse.

Table 6: Top 10 most frequent subjects and bookshelf topics in Chronoberg. Proportion of books assigned to each category are shown in % relative to the total number of books with at least one entry in the same category. "Subjects" are derived from Library of Congress headings, while "Bookshelves" are mostly from automatically assigned "browsing" categories. Note that multiple subject headings and bookshelves may be assigned to a single book.

| *Subjects* | | |
|---|---|---|
| Subject | Books | % |
| Fiction | 5998 | 23.9 |
| History | 3012 | 12.0 |
| Juvenile fiction | 2368 | 9.4 |
| Social Life and Customs | 1450 | 5.8 |
| 19th century | 1218 | 4.9 |
| England | 1154 | 4.6 |
| United States | 1052 | 4.2 |
| Great Britain | 872 | 3.5 |
| Description and Travel | 830 | 3.3 |
| Conduct of Life | 695 | 2.8 |

| *Bookshelves* | | |
|---|---|---|
| Bookshelf | Books | % |
| American Bestsellers 1895-1923 | 308 | 7.2 |
| Science Fiction | 285 | 6.7 |
| Children's Fiction 1895-1923 | 282 | 6.6 |
| Children's Series 1895-1923 | 207 | 4.9 |
| Children's Literature | 203 | 4.8 |
| World War I | 196 | 4.6 |
| US Civil War | 192 | 4.5 |
| Historical Fiction | 186 | 4.4 |
| Humor | 100 | 2.4 |
| Native America | 98 | 2.3 |

With these filtering steps, we successfully annotated 25,061 for chronological sorting out of the 73,500 books available in Project Gutenberg.

**Topic Distribution** The distribution of books and tokens in Chronoberg across decades is uneven, as illustrated in Figure 6, reflecting the availability of texts in Project Gutenberg. The number of books increases steadily up to 1920s, a trend likely fuelled by population growth, educational expansion, and economic development. A sharp decline follows, primarily due to copyright restrictions, with most texts published post-1929 remaining under copyright, and consequently being unavailable in Project Gutenberg. As a result, early 20th-century works are strongly represented, while the post-1920 are under-represented. This is especially evident for the 2000s, which contain only the year 2000, producing a notably low count of books and tokens for that decade.

Beyond temporal coverage, Project Gutenberg metadata also provides insights into Chronoberg's thematic composition. Table 6 presents the ten most common subjects and bookshelf topics, reflecting the historical context in which the books were written. Fiction, in its various forms, is the most prominent genre in Chronoberg. Many works also reflect social and historical contexts followed by works addressing social issues and major historical events such as World War I and U.S. Civil War. Historical works covering earlier periods are also well-represented. In contrast, children's literature

| **1750–1799** | |
| --- | --- |
| Author | Books |
| Gibbon, Edward | 14 |
| Schiller, Friedrich | 7 |
| Paine, Thomas | 7 |
| Wollstonecraft, Mary | 7 |
| Stanhope, Philip | 7 |

| **1800–1849** | |
| --- | --- |
| Author | Books |
| Bulwer-Lytton, Edward | 88 |
| Scott, Walter | 44 |
| Marryat, Frederick | 40 |
| Dickens, Charles | 39 |
| Cooper, James Fenimore | 33 |

| **1850–1899** | |
| --- | --- |
| Author | Books |
| Twain, Mark | 104 |
| Ballantyne, Robert Michael | 88 |
| Henty, George Alfred | 74 |
| Alger, Horatio Jr. | 72 |
| Fenn, George Manville | 66 |

| **1900–1949** | |
| --- | --- |
| Author | Books |
| Baum, Lyman Frank | 73 |
| Stratemeyer, Edward | 59 |
| Barbour, Ralph Henry | 57 |
| Oppenheim, Edward Phillips | 55 |
| Wells, Carolyn | 54 |

| **1950–2000** | |
| --- | --- |
| Author | Books |
| Leinster, Murray | 17 |
| Duellman, William Edward | 11 |
| Dick, Philip K. | 10 |
| Kjelgaard, Jim | 10 |
| Norton, Andre | 10 |

| **All time** | |
| --- | --- |
| Author | Books |
| Twain, Mark | 118 |
| Bulwer-Lytton, Edward | 104 |
| Ballantyne, Robert Michael | 88 |
| Fenn, George Manville | 87 |
| Henty, George Alfred | 86 |

Table 7: Chronoberg contains more than 20000 authors, out of which the most represented ones are shown and grouped by historical period (1750–2000). Book counts indicate the number of works attributed to each author within the dataset and illustrate shifts in author prominence over time.

constitutes a small portion of the dataset: the three bookshelf categories related to children's literature account for only 692 books, forming a relatively niche subject.

The subjects used to categorize the books are drawn from the Library of Congress Subject Headings, whereas the categories for bookshelves are derived from a mixture of hand-curated eBook collections and automatically assigned "browsing" categories. Multiple subject headings or bookshelves can be assigned to a single book. The completeness of the metadata varies, where only 22 books in Chronoberg lack subject headings, but around 83% not having any bookshelf assigned. These were excluded from the counts shown in Table 6.

We also examine the authors represented in CHRONOBERG over time. Out of the 20k authors in Chronoberg, the most represented authors are shown in Table 7, illustrating shifts in author prominence across 250 years. While several prolific writers (e.g., Dickens, Schiller, Twain) appear prominently, the corpus is largely shaped by authors who were popular in their time, but are rarely read today. This distribution reflects typical biases of large-scale digitized historical corpora, capturing the broader landscape of historical print culture, including many commercially popular authors whose work may better represent everyday linguistic usage of their period.

## A.3 INVESTIGATING THE EFFECT OF THE NUMBER OF NEIGHBORS ON VAD MEASURES

As discussed in Section 3.2 of the main text, the number of neighbours used to compute lexical scores is a crucial hyperparameter when constructing temporally aligned VAD lexicons. This choice directly affects the resulting VAD scores, where too few neighbours can introduce strong biases, while too

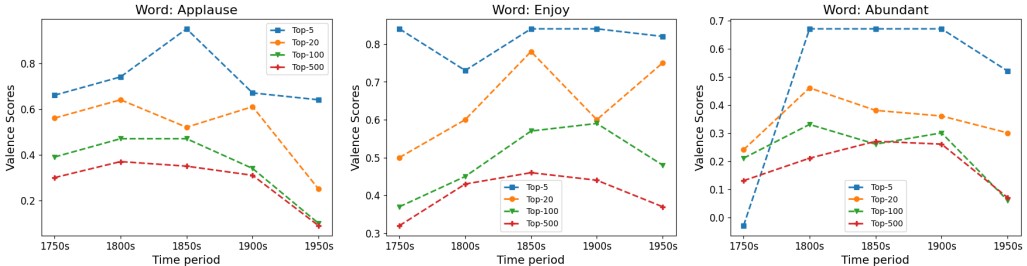

Figure 7: Visualizing the variance in the valence scores across the different time intervals. We vary the number of the Top-K neighbors to compute the affective valence scores for each word. Top-5 neighbors lead to strong bias, whereas top-500 neighbors lead to neutrality.

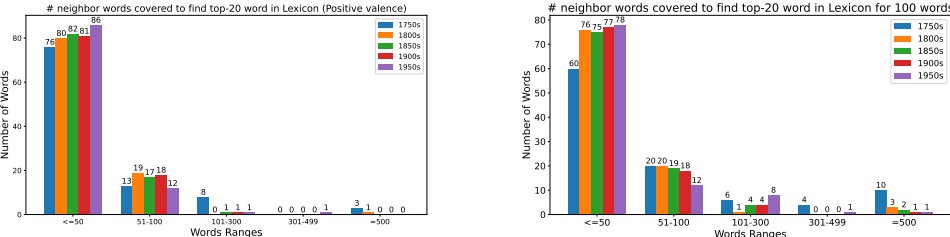

Figure 8: Visualizing the number of neighbors needs to be traversed to determine a VAD score by identifying the top-20 words in NRC VAD Lexicons. For most words, top-50 neighbors are sufficient while top-100 nearest neighbors provide a reliable upper bound covering all words.

many can produce overly neutral scores. Figure 7 illustrates this effect across the different eras for a select set of examples.

From Figure 7 it is observable that using the top-20 neighbours is typically adequate. However, because Chronoberg's lexicons are based on the contemporary NRC VAD lexicon, not all top-20 neighbours in historical sentences may have corresponding NRC VAD scores. To address this, we conducted a small experiment, where we randomly sampled 100 words and measured how many nearest neighbours must be traversed to identify the 20 words present in the NRC VAD lexicon. The results in Figure 8 show that for most words, considering the top 50 neighbours is sufficient to obtain 20 valid scores, and the top 100 neighbours provide a reliable upper bound covering all words. Based on this analysis, we adopt the top-100 neighbours as the standard for computing VAD scores in Chronoberg, ensuring robust and consistent affective annotations.

### A.4 FURTHER EXAMPLES FOR SEMANTIC SHIFTS OF WORDS IN CHRONOBERG

We present more qualitative examples that illustrate the diachronic transition of affective connotations, both from positive to negative and vice versa in Table 8. For instance, sanctimonious, which once conveyed genuine holiness, has shifted to denote a hypocritical display of moral superiority. Likewise, weird, originally associated with the supernatural and unearthly, now predominantly means "odd," "strange," or "bizarre." Another compelling case is depressive, which evolved from a neutral or even positive association of "pressing down" to a word carrying strongly negative emotional connotations. Gay is another example where we see a definite transition in the valence score from being an overtly positive word (0.70) to a more neutral word (0.15).

Conversely, we also observe cases of semantic shift from negative to positive. Words such as *infatuation, destiny, tweak, and repertoire* exemplify this trend. Infatuation has moved from its earlier sense of "making foolish" to its modern meaning of intense admiration. Similarly, tweak, once meaning "to pluck or pinch," has broadened to signify the act of making small adjustments. We have also reported several instances of words that were predominantly positive or negative across the time interval of 250 years, as shown in Table 9.

Table 8: Temporal change in valence scores across centuries. We compute the top-20 neighbours of some negative words from the Lexicon and took the mean to obtain the individual scores

| Words | *Positive → Negative* | | | | | Words | *Negative → Positive* | | | | |
|---|---|---|---|---|---|---|---|---|---|---|---|
| | 1750s | 1800s | 1850s | 1900s | 1950s | | 1750s | 1800s | 1850s | 1900s | 1950s |
| asylum | 0.27 | -0.24 | -0.54 | -0.52 | -0.76 | bloomers | 0.01 | -0.05 | 0.27 | 0.18 | 0.66 |
| coronary | 0.17 | -0.13 | -0.22 | -0.15 | -0.55 | destiny | -0.54 | 0.06 | 0.32 | 0.11 | 0.44 |
| depressive | 0.3 | -0.96 | -0.56 | -0.65 | -0.74 | dunk | 0.42 | 0.15 | None | -0.18 | 0.35 |
| germs | 0.15 | 0.26 | -0.14 | -0.55 | -0.68 | febrile | -0.58 | -0.53 | -0.66 | -0.54 | 0.06 |
| heartbreak | 0.18 | -0.6 | -0.69 | -0.74 | -0.81 | infatuation | -0.66 | -0.63 | -0.52 | -0.35 | 0.53 |
| homeless | 0.11 | -0.62 | -0.66 | -0.63 | -0.28 | karma | 0.04 | None | 0.25 | 0.14 | 0.32 |
| malfeasance | 0.27 | -0.56 | -0.48 | -0.65 | -0.72 | outing | 0.67 | -0.22 | 0.58 | 0.58 | 0.57 |
| punk | 0.2 | 0.14 | -0.25 | -0.17 | -0.26 | repertoire | -0.65 | 0.32 | 0.38 | 0.4 | 0.39 |
| sanctimonious | 0.11 | -0.14 | -0.37 | -0.57 | -0.81 | sanitation | 0.28 | -0.06 | -0.01 | 0.01 | 0.32 |
| senile | None | -0.52 | -0.56 | -0.69 | -0.74 | stockbroker | 0.02 | 0.15 | 0.08 | 0.16 | 0.34 |
| weird | 0.3 | 0.01 | -0.28 | -0.33 | -0.49 | technology | None | 0.06 | 0.14 | 0.18 | 0.38 |
| jolly | 0.05 | 0.42 | 0.49 | 0.56 | -0.43 | tweak | 0.04 | -0.19 | -0.19 | -0.12 | 0.67 |

Table 9: Temporal change in valence scores across centuries. We compute the top-20 neighbors of some negative words from the Lexicon and took the mean to obtain the individual scores

| Words | *Positive Words* | | | | | Words | *Negative Words* | | | | |
|---|---|---|---|---|---|---|---|---|---|---|---|
| | 1750s | 1800s | 1850s | 1900s | 1950s | | 1750s | 1800s | 1850s | 1900s | 1950s |
| abundant | 0.28 | 0.37 | 0.4 | 0.4 | 0.41 | afraid | -0.35 | -0.34 | -0.09 | -0.27 | -0.39 |
| enjoy | 0.52 | 0.6 | 0.62 | 0.63 | 0.86 | angered | -0.2 | -0.58 | -0.66 | -0.66 | -0.69 |
| hugs | -0.28 | 0.34 | 0.56 | 0.56 | 0.4 | annihilation | -0.49 | -0.68 | -0.62 | -0.73 | -0.61 |
| laughter | 0.08 | 0.33 | 0.49 | 0.49 | 0.75 | bankruptcy | -0.33 | -0.37 | -0.46 | -0.55 | -0.46 |
| liking | 0.35 | 0.07 | 0.12 | 0.28 | 0.21 | betray | -0.52 | -0.44 | -0.51 | -0.49 | -0.66 |
| lucky | -0.14 | 0.05 | 0.33 | 0.3 | 0.29 | chaos | -0.27 | -0.33 | -0.23 | -0.46 | -0.57 |
| marvel | 0.03 | 0.56 | 0.63 | 0.77 | 0.84 | stabbed | -0.48 | -0.68 | -0.74 | -0.6 | -0.56 |
| merry | 0.56 | 0.71 | 0.76 | 0.79 | 0.54 | strangulation | -0.6 | -0.47 | -0.67 | -0.69 | -0.26 |
| respectful | 0.45 | 0.42 | 0.41 | 0.4 | 0.62 | suicidal | -0.27 | -0.75 | -0.7 | -0.71 | -0.59 |

# B  ANALYSIS OF HATE SPEECH AND HARMFUL LANGUAGE

This appendix section provides further information on how suitable hate-detection tools were identified and benchmarked, as well as how they were applied to Chronoberg, and includes additional examples.

Table 10: We also report changes in valence scores derived from lexicons built using Word2Vec models trained on 20-year temporal slices. When comparing these results to the scores obtained from models trained on 50-year slices, we observe a consistent pattern: words exhibiting a negative-to-positive shift in the 50-year models also show comparable transitions in the 20-year models. However, the 20-year splits reveal a finer-grained progression in the valence scores compared to the coarser counterpart.

| Words | *Positive → Negative* | | | | | | | | | | | | |
|---|---|---|---|---|---|---|---|---|---|---|---|---|---|
| | 1750s | 1770s | 1790s | 1810s | 1830s | 1850s | 1870s | 1890s | 1910s | 1930s | 1950s | 1970s | 1990s |
| asylum | 0.16 | 0.06 | 0.16 | -0.01 | -0.2 | -0.5 | -0.49 | -0.54 | -0.52 | -0.54 | -0.46 | -0.1 | -0.1 |
| germs | 0.3 | 0.28 | 0.21 | 0.28 | 0.31 | 0.17 | 0.15 | -0.54 | -0.52 | -0.68 | -0.53 | -0.53 | -0.06 |
| homeless | 0.2 | -0.23 | -0.44 | -0.53 | -0.66 | -0.65 | -0.64 | -0.66 | -0.63 | -0.49 | -0.52 | -0.52 | -0.2 |
| punk | 0.17 | -0.31 | -0.16 | -0.16 | -0.16 | -0.21 | -0.21 | -0.16 | -0.12 | -0.35 | -0.14 | -0.11 | -0.11 |
| weird | 0.2 | 0.2 | 0.05 | -0.11 | -0.0 | -0.32 | -0.29 | -0.26 | -0.36 | -0.47 | -0.44 | -0.17 | -0.17 |

Table 11: Performances of models on full HateCheck test set. The best values are indicated in bold.

| Model | Acc | F1 | P | R |
|-------|-----|-----|-----|-----|
| Perspective API (Lees et al., 2022) | 0.578 | 0.559 | **0.993** | 0.389 |
| pysentimiento (Pérez et al., 2024) | 0.521 | 0.527 | 0.820 | 0.388 |
| Facebook's RoBERTa (Liu et al., 2019) | **0.956** | **0.968** | 0.963 | **0.973** |
| English Abusive MuRIL (Das et al., 2022) | 0.491 | 0.558 | 0.694 | 0.466 |
| BERT HateXplain (Devlin et al., 2019) | 0.384 | 0.270 | 0.730 | 0.165 |
| DehateBERT Mono English (Aluru et al., 2020) | 0.425 | 0.351 | 0.784 | 0.226 |
| IMSyPP Hate Speech (Kralj Novak et al., 2022) | 0.750 | 0.826 | 0.790 | 0.866 |
| Twitter RoBERTa Large Hate (Antypas et al., 2023) | 0.615 | 0.640 | 0.898 | 0.497 |
| DistilRoBERTa Hateful Speech (Hugging Face, 2023) | 0.568 | 0.652 | 0.730 | 0.590 |

Table 12: Perspective API scores for ChronoBerg sentences labelled hateful by the RoBERTa model

(a) Distribution and precision of hateful sentences across different score intervals

| Score $\in$ | Sentences | TP | FP |
|-------------|-----------|-----|-----|
| [0.0, 0.1) | 2,411,275 | - | - |
| [0.1, 0.2) | 360,085 | - | - |
| [0.2, 0.3) | 228,128 | - | - |
| [0.3, 0.4) | 148,038 | - | - |
| [0.4, 0.5) | 99,557 | - | - |
| [0.5, 0.6) | 68,540 | 84 | 16 |
| [0.6, 0.7) | 22,213 | 95 | 5 |
| [0.7, 0.8) | 5,116 | 99 | 1 |
| [0.8, 0.9) | 470 | 100 | 0 |
| [0.9, 1.0] | 11 | 11 | 0 |

(b) Precision and size of the set of hateful sentences depending on threshold choice

| Score $\geq$ | Sentences | P |
|--------------|-----------|-----|
| 0.0 | 3,343,433 | - |
| 0.1 | 932,158 | - |
| 0.2 | 572,073 | - |
| 0.3 | 343,945 | - |
| 0.4 | 195,907 | - |
| 0.5 | 127,366 | 87.4% |
| 0.6 | 27,710 | 95.8% |
| 0.7 | 5,597 | 99.1% |
| 0.8 | 481 | 100% |
| 0.9 | 11 | 100% |

## B.1 IDENTIFYING SUITABLE HATE-DETECTION TOOLS WITH HATECHECK

We provide further insights into the choices underlying the main body's hate-speech detection pipeline to contextualize harmful language in Chronoberg. In total, we have considered nine different modern hate speech detection tools: Pysentimiento toolkit (Pérez et al., 2024), Google Perspective API (Lees et al., 2022), Facebook RoBERTa (Liu et al., 2019), as well as 7 most popular Hugging Face models when filtering for the keyword "hate". It is worth noting that all nine tools are built upon one of the two popular transformer architectures BERT (Devlin et al., 2019) and RoBERTa (Liu et al., 2019).

Consequently, we evaluated these tools on the HateCheck benchmark (Röttger et al., 2021), a suite of functional tests for hate speech detection. As shown in Table 11, many existing hate speech detection tools performed no better than random guessing. RoBERTa (Liu et al., 2019) was the only tool that stood out, demonstrating consistently strong performance, achieving high recall, while the remaining tools exhibited substantial limitations in either recall, precision, or overall reliability. Notably, Perspective API (Lees et al., 2022) achieved exceptionally high precision, making it particularly viable for curating a subset of potentially hateful sentences, despite its limited coverage.

To combine their strengths, we adopted a two-stage pipeline in the main body based on these HateCheck observations. First, we use RoBERTa (Liu et al., 2019) to flag a broad set of potentially hateful sentences. These sentences are then filtered by the Perspective API (Lees et al., 2022) to reduce false positives. This approach balances scalability and precision, addressing RoBERTa's over-sensitivity and the Perspective API's limited coverage.

## B.2 SENTENCES CONTAINING POTENTIAL HATE IN CHRONOBERG

Table 12 presents a detailed analysis of Perspective API scores for sentences flagged as hateful by the RoBERTa model for Chronoberg. In Table 12(a), the distribution of these sentences across score intervals is shown, along with manual annotations of 100 sampled sentences per range to estimate the

precision. No manual revision was conducted for scores below 0.5, as the [0.5, 0.6) range already yielded 16 false positives, indicating a substantial drop in precision. In consequence, lower thresholds seem to be impractical for reliable hate speech filtering.

Table 12(b) complements this by showing how varying the Perspective API threshold affects both the number of flagged sentences and the estimated precision. We extrapolated precision estimates from the interval-level annotations, due to the infeasibility of reviewing all 3.3 million samples. Notably, 2.4 million sentences, making up 72.1%, that were labeled as hateful by RoBERTa scored below 0.1 in Perspective API, indicating a high false positive rate. Despite this, RoBERTa exhibited strong recall during evaluation and likely also captured a majority of hateful sentences, albeit imprecisely. Using a threshold of 0.7 with Perspective API resulted in a highly precise subset of 5,597 sentences.

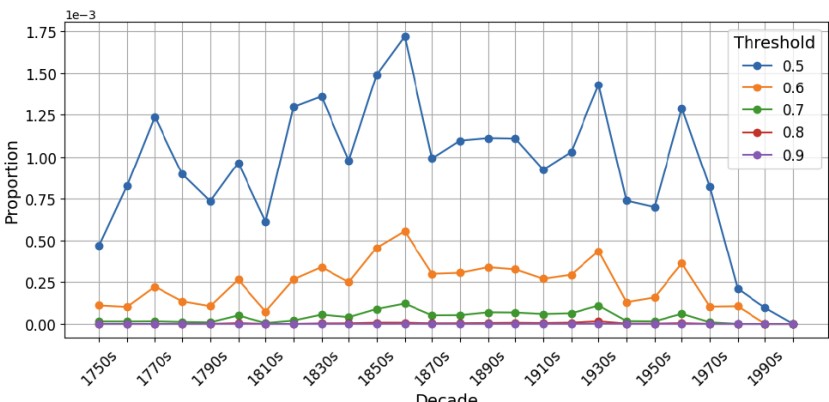

Figure 9: Proportion of hateful sentences over time for different Perspective API thresholds

We observed that while varying the Perspective API threshold influences the volume of hate detected across decades, as seen in Figure 9, the overall temporal patterns remain similar and are retained. In comparison with the distribution as flagged by RoBERTa, more substantial differences are revealed, particularly in earlier historical periods, whereas Perspective API consistently detects less hate across all thresholds.

### B.3 FURTHER QUALITATIVE EXAMPLES OF HARMFUL LANGUAGE ANALYSIS

In complement to the examples shown in section 4 of the main body, we present additional examples examining the alignment between hate-check outputs from LLMs and VAD scores in Table 13. We observe comparable trends and discrepancies in sentiment classification, particularly in how modern classifiers often fail to recognize historically situated expressions of hate or, conversely, overgeneralize from present-day connotations.

## C SEQUENTIAL TRAINING AND ADAPTATION OF LLMs ON CHRONOBERG

This section outlines the training configuration, the continual learning variants chosen, and further insights from evaluating temporal generalization in Chronoberg.

### C.1 TRAINING CONFIGURATION AND CONTINUAL LEARNING VARIANTS

All experiments were carried out with the Pythia 1.4B architecture (Biderman et al., 2023), using the gpt-neo-1.3B tokenizer. Models were trained on NVIDIA A100-80GB GPUs for 30 epochs with a batch size of 64 and a micro batch size of 4, using gradient accumulation of 8. The optimizer was Adam (Kingma & Ba, 2014) with a weight decay of 0.1. The optimizer's $\beta_1$ parameter were set to 0.9 and $\beta_2$ was set to 0.95. The learning rate was $1 \times 10^{-4}$, and the learning rate scheduler followed a cosine decay schedule. The warmup strategy was linear, with 100 warmup steps. The sequence length was set to 2048 tokens.

Table 13: Extended illustrative examples. Comparison of sentence-level classifications in Chronoberg across time intervals using LLM-based hate-check tools (RoBERTa+Perspective API, OpenAI) (🚩- Hate, ✓- Non-hate) and valence-based scoring (👎- Negative, 👍- Positive sentiment). We grouped the instances based on model agreement. The first group of rows illustrates cases where all tools collectively classify an instance as harmful. The group of rows 2-3 shows instances where they disagree, classifying them as either positive or negative. While VAD lexicons provide interpretable complementary signals, we recognize that harmful texts are inherently subjective; thus, we do not regard them as definitive solutions to LLM misclassification but as potential tools to enhance LLM performance.

| YEAR | Sentences | Hate-Check Models RoBERTA+Persp | OpenAI | Valence Score | Affective connotation |
|---|---|---|---|---|---|
| 1750s | but i loathe you, you apache indian! | 🚩 | 🚩 | -0.48 | 👎 |
| 1800s | we has slaves too; we has niggers to a stand-still. | 🚩 | 🚩 | -0.37 | 👎 |
| 1850s | the irish are mostly very filthy and diseased. | 🚩 | 🚩 | -0.51 | 👎 |
| 1850s | i hate women. | 🚩 | 🚩 | -0.40 | 👎 |
| 1900s | you never want to take a nigger into your conferences. | 🚩 | 🚩 | -0.18 | 👎 |
| 1900s | kill every black bastard befo mornin! | 🚩 | 🚩 | -0.365 | 👎 |
| 1900s | The Bhutanese women are the ugliest specimens of femininity I have ever seen. | 🚩 | 🚩 | 0.45 | 👎 |
| 1950s | i hate chicago, i hate americans! | 🚩 | 🚩 | -0.40 | 👎 |
| 1950s | i hate the germans! | 🚩 | 🚩 | -0.40 | 👎 |
| 1750s | A man may play with decency; but if he games, he is disgraced. | ✓ | ✓ | -0.68 | 👎 |
| 1750s | Defamation and calumny never attack, where there is no weak place; they magnify, but they do not create. | ✓ | ✓ | -0.76 | 👎 |
| 1750s | thou traitor, hie away; By all my stars I thou enviest Tom Thumb | ✓ | 🚩 | -0.69 | 👎 |
| 1800s | he redoubled his gayety and carelessness. | ✓ | ✓ | -0.69 | 👎 |
| 1800s | who the beggar was that i killed | ✓ | ✓ | -0.60 | 👎 |
| 1800s | what hatred she distills! | ✓ | ✓ | -0.72 | 👎 |
| 1850s | The piece was stupid beyond expression | ✓ | ✓ | -0.57 | 👎 |
| 1950s | so it is a hell of women, is it? | ✓ | ✓ | -0.35 | 👎 |
| 1750s | The conversation at supper was very gay. | 🚩 | ✓ | 0.37 | 👍 |
| 1750s | In my way home to my tent, I saw a faggot lying in the way, | 🚩 | 🚩 | 0.05 | 👍 |
| 1850s | Religion prescribes obedience. | 🚩 | ✓ | 0.08 | 👍 |
| 1850s | Where is the woman to strew the flowers? | 🚩 | 🚩 | 0.05 | 👍 |
| 1900s | I may cut you out of my gold expedition, if you get gay. | 🚩 | 🚩 | 0.06 | 👍 |

We explored several training regimes to evaluate how models adapt to temporal shifts in language and semantics:

- **Sequential Training:** The model is trained incrementally on consecutive 50-year time intervals of Chronoberg. Each interval updates the weights sequentially, simulating long-term continual learning. This setup provides a baseline to measure the impact of catastrophic forgetting and the ability of the model to retain knowledge from earlier time periods.

- **Single-Interval Baseline:** Independent models are trained from scratch on each 50-year time interval. This setup isolates temporal intervals, allowing us to assess the model's performance on temporally localized data without interference from other periods. This setting serves as a control to evaluate how well a model can learn within a single time window.

- **Elastic Weight Consolidation (EWC) (Kirkpatrick et al., 2017):** EWC adds a regularization term based on the Fisher Information Matrix to penalize changes to parameters critical for previously learned experiences. After training on a given time interval, the Fisher Information is computed for all parameters and subsequent updates are constrained, controlled by a regularization strength. This method mitigates catastrophic forgetting while allowing adaptation to new time intervals.

- **Low-Rank Adaptation (LoRA) (Hu et al., 2022):** LoRA injects trainable low-rank matrices into the attention layers of the model, allowing efficient adaptation with a small number of parameters. For our experiments, we set the rank $r = 8$ and scaling factor $\alpha = 16$. This method allows flexible learning for new time intervals, while preserving the frozen base model, offering a trade-off between retention of old knowledge and the ability to learn from new data (i.e. plasticity).

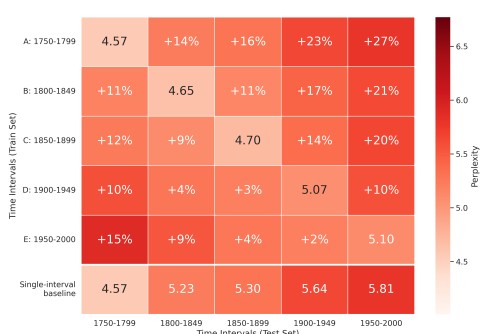 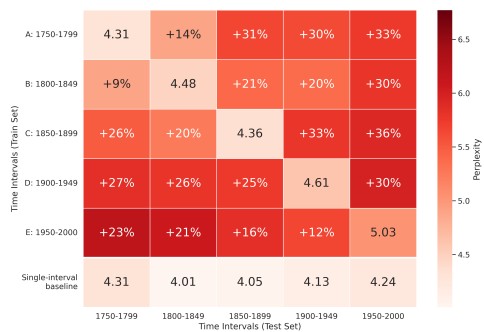

(a) Perplexity for valence-stable words.      (b) Perplexity for valence-shifting words.

Figure 10: Perplexity of continually trained models with LoRA adapters, evaluated on test sets with words that (a) remain stable in valence, and (b) exhibit valence shift. Higher perplexity indicates worse language modelling performance. LoRA maintains low perplexity on valence-stable words, preserving diagonal performance and reducing catastrophic forgetting (eg., only a 15% rise for the initial interval at the end). At the same time, LoRA offers greater plasticity. However, as with other approaches, perplexities increase more sharply for valence-shifting words (e.g., the model at time interval E: 1950-2000 shows a +23% rise when evaluated on earlier intervals), highlighting the persistent difficulty of consolidating semantic shifts over time.

## C.2 LoRA Results and Additional Experimental Result Discussion

This section complements the experimental results of Section 4.1 in the main body. Recall that we examine how language models trained under different temporal regimes capture semantic shifts over time, focusing on words whose affective meaning either changes or remains stable.

Our main results compared sequential fine-tuning (ST) and Elastic Weight Consolidation (EWC). ST showed gradual degradation over time and severe forgetting of earlier intervals, with particularly high perplexity on valence-shifting words. EWC mitigated forgetting more effectively for valence-stable words, keeping perplexities closer to diagonal performance. However, it also remained limited in forward generalization, struggling with valence-shifting words, where semantic drift impeded consolidation.

Notably, the patterns observed with ST and EWC are in complete agreement with results obtained using Low-Rank Adaptation (LoRA), shown in this section of the appendix. By introducing trainable low-rank matrices on top of frozen model weights, LoRA enables flexible adaptation to new intervals while retaining prior knowledge. As seen in Figure 10a, LoRA maintains diagonal perplexities close to the single-interval baseline, indicating strong retention of knowledge over time. Off-diagonal values show moderate increases relative to diagonal values, such as a 15% rise for the model at time interval E (1950-2000) when tested on earlier time intervals. Forward generalization (top-right triangle values) remain challenging, yet the degradation is milder than in the other two cases.

In the valence-shifting setting (Figure 10b), LoRA again follows the same qualitative pattern as previously discussed. Localized diagonal performance remains intact, but cross-temporal perplexities rise substantially (e.g., for model at time interval E increases 23% on earlier intervals). Forward generalization is hindered by semantic drift. As hypothesized, this demonstrates the inherent challenge of consolidating contradictory affective meanings across time.

We additionally provide experimental results for Pythia 160M in Figure 11. The model's relatively limited capacity amplifies the trends we observe, since small models have fewer parameters to store and consolidate knowledge, making them more susceptible to catastrophic forgetting under sequential training (Figure 11a) and more sensitive to semantic drift over time (eg., 30% rise for the model at interval E:1950-2000 in Figure 11b). EWC mitigates forgetting more effectively than ST, reducing off-diagonal perplexity increase (eg., 25% rise for valence-stable words in Figure 11c vs. 27% rise in Figure 11a), but forward generalization remains challenging. LoRA, in contrast, retains diagonal performance while offering improved plasticity for future intervals, resulting in lower perplexities overall(for model at time interval E(1950-2000) in Figure 11e, we observe only a 23%

Table 14: Comparison of continual learning strategies on Chronoberg: The values show average perplexity increase over time for the Pythia 1.4b model. (Perplexity), perplexity increase when generalizing to unseen future intervals (Forward Gen.), and best/worst case perplexities (excluding same-interval evaluations in the diagonal) across all time intervals. Sequential fine-tuning suffers most from forgetting and generalization errors, EWC reduces forgetting but struggles with semantic shifts, while LoRA offers an intermediate trade-off between retention and adaptability.

| Method | Perplexity | Forward Gen. | Best Case | Worst Case |
|---|---|---|---|---|
| Sequential FT | 34% ↑ | 33% ↑ | 4.58 (1900–49) | 6.64 (1950–2000) |
| EWC | 12% ↑ | 29% ↑ | 4.65 (1850–99) | 6.77 (1950–2000) |
| LoRA | 15% ↑ | 27% ↑ | 4.48 (1800–49) | 6.19 (1950–2000) |

rise in perplexity and +26% in Figure 11f). Although the overall perplexities remain elevated due to the small model size, the qualitative patterns such as strong diagonal retention, moderate off-diagonal increases, and greater difficulty with semantic drift remain consistent. This further demonstrates that the conflicting nature of semantically shifting words hinders even continual learning methods from fully consolidating past knowledge.

Overall, our results highlight a clear trade-off between knowledge retention and adaptability to semantic change across the three sequential learning strategies. Table 14 summarizes these patterns for Pythia 1.4b, illustrating how each method balances retention and adaptability in capturing semantic drift. Sequential fine-tuning suffers most from forgetting, particularly for valency-shifting words, while EWC preserves prior knowledge effectively but simultaneously constrains future learning (i.e. plasticity). LoRA offers an intermediate solution, retaining knowledge nearly as well as EWC while providing more plasticity for future time intervals, resulting in lower perplexity on the current time interval (values on the diagonal) than ST and slightly higher than EWC.

Our findings demonstrate that simple temporal adaptation is insufficient for exposure of models to the natural temporal flow of language, as captured by Chronoberg. This is particularly relevant for socially or affectively pertinent terms whose meaning have evolved.

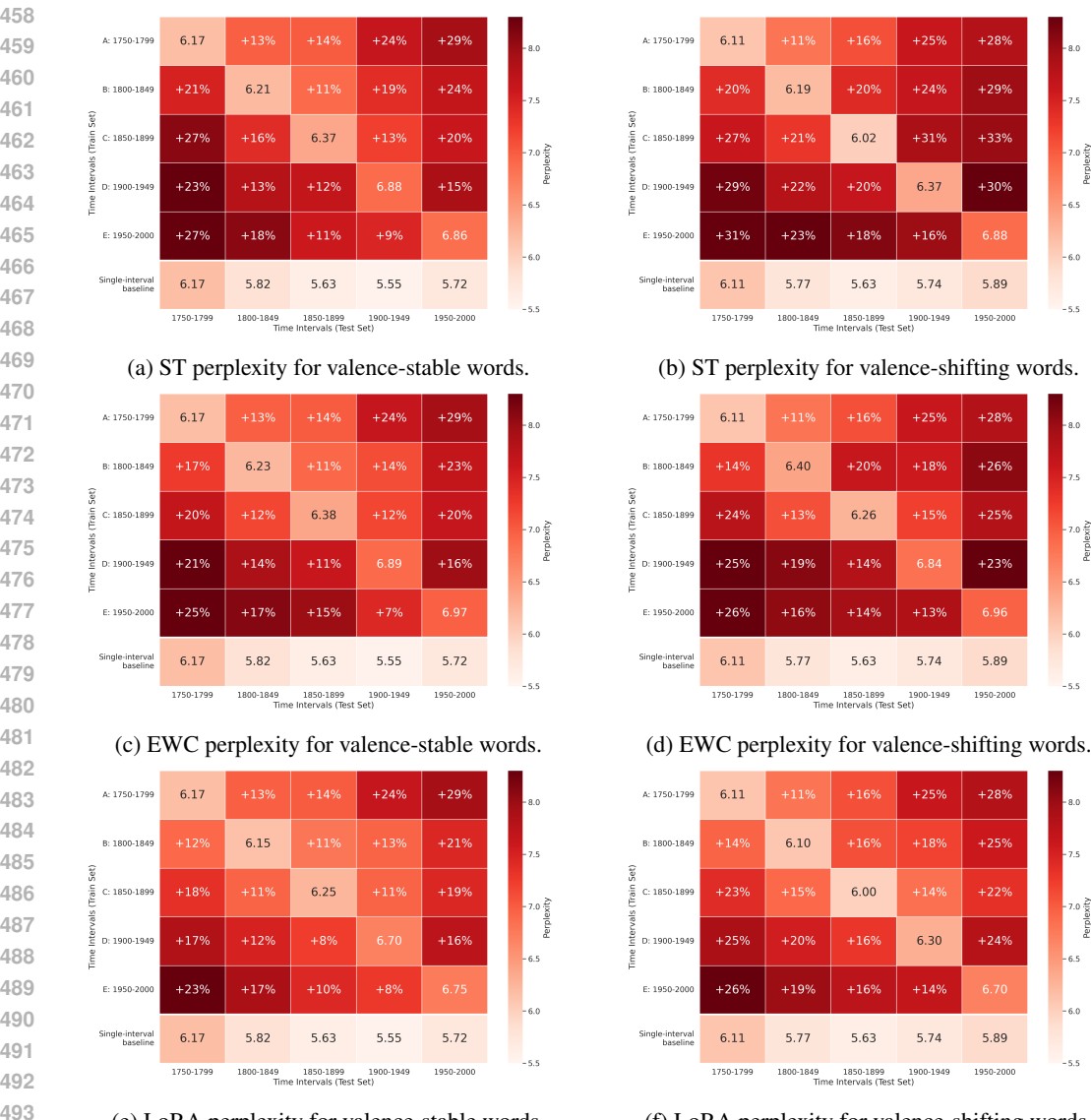

(a) ST perplexity for valence-stable words.

(b) ST perplexity for valence-shifting words.

(c) EWC perplexity for valence-stable words.

(d) EWC perplexity for valence-shifting words.

(e) LoRA perplexity for valence-stable words.

(f) LoRA perplexity for valence-shifting words.

Figure 11: Perplexity of Pythia-160m models trained sequentially(ST), with EWC and LoRA adapters, evaluated on test sets with words that (a,c,e) remain stable in valence, and (b,d,f) exhibit valence shift. Higher perplexity indicates worse language modelling performance. We observe that sequentially trained models suffer from catastrophic forgetting (lower-left off-diagonal) more strongly on valence-stable words than valence-shifting ones. For instance, backward evaluation of the last interval (row E:1950-2000) on the initial interval shows a +27% rise for valence-stable words compared to +31% for valence shifting words. Forward generalization (upper-right off diagonal) is limited, particularly in later intervals. EWC reduces catastrophic forgetting (eg., 26% rise vs. 31% for ST on the initial interval) but forward generalization remains overall challenging. The benefit is more prominent for valence-stable words where semantic shifts are easier to consolidate. LoRA maintains low perplexity on valence-stable words, preserving diagonal performance and reducing catastrophic forgetting (+23% rise for initial interval) while offering improved forward generalization (+24% rise). However, as with other approaches, valence-shifting words still show increase in perplexity, highlighting persistent difficulty of consolidating knowledge and generalization.

## D CHRONOBERG DATASHEET

| Motivation |
|:---:|

**For what purpose was the dataset created?** (e.g. was there a specific task in mind? Was there a specific gap that needed to be filled? Please provide a description.)

The Chronoberg dataset was created to provide a temporally structured corpus to support large-scale language modelling and linguistic analysis over time. While existing resources provide broad coverage, they typically lack long-term temporal structure, and are not well-suited to studying semantic drift and diachronic variation. Chronoberg was designed to support tasks such as:

- Sequential training and continual learning of LLMs across time,
- Evaluation of temporal generalization and (catastrophic) forgetting,
- Construction of historically grounded affective lexicons for systematic linguistic and affective analysis, and
- Analysis of detection of discriminatory and sensitive language in historical contexts.

**What (other) tasks could the dataset be used for?** Are there obvious tasks for which it should not be used?

The dataset was created to provide a scalable benchmark for tasks such as:

- Sequential adaptation of LLMs across centuries,
- Concept drift modelling and continual learning,
- Editing and unlearning to modify or update interpretations of certain English words or sentences.

The VAD score annotations in the dataset are not intended for tasks that require absolute semantic evaluation, but study relative semantic change.

**Who created this dataset (e.g., which team, research group) and on behalf of which entity ?** (e.g., company, institution, organization)

Anonymized for review

**Who funded the creation of the dataset?** If there is an associated grant, please provide the name of the grantor and the grant name and number.

Anonymized for review

**Any other comments?**

No further comments.

| Composition |
|:---:|

**What do the instances that comprise the dataset represent (e.g., documents, photos, people, countries)?** Are there multiple types of instances (e.g., movies, users, and ratings; people and interactions between them; nodes and edges)? Please provide a description.

The instances in the Chronoberg dataset represent digitized books sourced from Project Gutenberg. Each instance corresponds to:

- Full text of a book
- Temporal metadata (publication year)
- Lexicons that capture affective sentiment dimensions such as Valence, Dominance, and Arousal.

- Sentence level annotations associating a VAD score.

**How many instances are there in total (of each type, if appropriate)?**

Chronoberg is composed of 2.7 billion tokens, representing roughly 91 million sentences from 25,061 English-language books published between 1750 and 2000, with additional metadata in the form of temporally-aligned VAD lexicons that span 337,458 words.

**Does the dataset contain all possible instances or is it a sample (not necessarily random) of instances from a larger set?** If the dataset is a sample, then what is the larger set? Is the sample representative of the larger set (e.g., geographic coverage)? If so, please describe how this representativeness was validated/verified. If it is not representative of the larger set, please describe why not (e.g., to cover a more diverse range of instances, because instances were withheld or unavailable).

Chronoberg is curated from Project Gutenberg, an openly accessible corpus of literary texts. Our primary focus was to derive a temporally structured dataset suitable for studying diachronic semantic drift, i.e., how word and sentence meanings evolve across centuries. To this end, we restricted our scope to Late Modern English works published between 1750 and 2000. Since Project Gutenberg's metadata on original publication date is often inaccurate or missing, we developed an inference pipeline that leverages the OpenLibrary API to obtain corrected publication years. After this process, we retained a total of 25,061 books.

**What data does each instance consist of? "Raw" data (e.g., unprocessed text or images) or features?** In either case, please provide a description.

Each instance represents processed texts. In addition, we have also introduced temporally-aligned VAD lexicons as part of our metadata, where each instance represents processed English words.

**Is there a label or target associated with each instance?** If so, please provide a description.

The dataset is grouped by year of publication. So, texts contain labels in the form of a specific year to which they belong. In addition, we have provided valence, arousal, and dominance annotations for each sentence in a specific time period.

**Is any information missing from individual instances?** If so, please provide a description, explaining why this information is missing (e.g., because it was unavailable). This does not include intentionally removed information, but might include, e.g., redacted text.

There is no information missing, as we have excluded such examples from the dataset in its construction process.

**Are relationships between individual instances made explicit (e.g., users' movie ratings, social network links)?** If so, please describe how these relationships are made explicit.

Relationships between the individual instances are made explicit through their shared temporal alignment. Each book instance is linked to a publication year, which allows for grouping, comparison and sequential ordering across time. Instances can also be grouped into genres such as Fiction, History, Social Life, Conduct of Life, and Travel, inherited from the original Project Gutenberg corpus. At the same time, given the historical span of 1750–2000, the texts could also be categorized with respect to major historical contexts and events of the period. We did not pursue this latter categorization in depth, as we believe it requires domain expertise beyond the scope of our work.

**Are there recommended data splits (e.g., training, development/validation, testing)?** If so, please provide a description of these splits, explaining the rationale behind them.

Yes. Although the best performing year predictor yields an uncertainty of 3-5 years, this margin is negligible at the scale of our diachronic analysis. To ensure robustness against temporal noise and to better capture semantic shifts over time, we recommend that alternately created training, validation and test sets be constructed within coarse temporal bins no smaller than 10-15 years.

**Are there any errors, sources of noise, or redundancies in the dataset?** If so, please provide a description.

Yes, as mentioned above there is an uncertainty of around 3-5 years as a consequence of our publication date inference pipeline. The uncertainty originates from the fact that publication dates are often missing or refer to the time of digitization in their original online repository and alternate sources needed to be inquired. We belief the uncertainty is acceptable in light of our considered 250 year time range.

**Is the dataset self-contained, or does it link to or otherwise rely on external resources (e.g., websites, tweets, other datasets)?** If it links to or relies on external resources, a) are there guarantees that they will exist, and remain constant, over time; b) are there official archival versions of the complete dataset (i.e., including the external resources as they existed at the time the dataset was created); c) are there any restrictions (e.g., licenses, fees) associated with any of the external resources that might apply to a future user? Please provide descriptions of all external resources and any restrictions associated with them, as well as links or other access points, as appropriate.

Yes, the dataset is self-contained. There are no access restrictions or required external resources.

**Does the dataset contain data that might be considered confidential (e.g., data that is protected by legal privilege or by doctor-patient confidentiality, data that includes the content of individuals non-public communications)?** If so, please provide a description.

No.

**Does the dataset contain data that, if viewed directly, might be offensive, insulting, threatening, or might otherwise cause anxiety?** If so, please describe why.

Yes. We acknowledge that the literary texts from the 1750s to the 2000s that compose Chronoberg may contain sensitive content that could be offensive, insulting, or threatening to certain groups. While we do not intend to objectify anyone, we also aim to preserve the integrity of the original works without alteration and avoiding historical erasure. At the same time, this approach opens up new possibilities for exploring how emotions were directed toward specific groups during different historical periods.

**Does the dataset relate to people?** If not, you may skip the remaining questions in this section.

Yes, we acknowledge that Chronoberg contains instances of text that may refer to groups of people or individuals, either directly or indirectly. As it comprises historical literary works from the 1750s to the 2000s, some texts are curated from literal biographies of historical figures, while others relate to wars and other significant events of the period.

**Does the dataset identify any subpopulations (e.g., by age, gender)?** If so, please describe how these subpopulations are identified and provide a description of their respective distributions within the dataset.

Yes, there are literary works that refer to or identify subpopulations by age, group or gender, considering that literary works span across different genres and include historical content.

**Is it possible to identify individuals (i.e., one or more natural persons), either directly or indirectly (i.e., in combination with other data) from the dataset?** If so, please describe how.

Yes, in particular historical figures are explicitly identified in books containing historical non-fiction content.

**Does the dataset contain data that might be considered sensitive in any way (e.g., data that reveals racial or ethnic origins, sexual orientations, religious beliefs, political opinions or union memberships, or locations; financial or health data; biometric or

**genetic data; forms of government identification, such as social security numbers; criminal history)?** If so, please provide a description.

Yes. While the dataset is composed of historical texts rather than personal records, it contains language that may be considered sensitive. This includes expressions of racial, ethnic, religious, gendered, political bias reflective of the time periods covered (1750-2000). Such content may involve discriminatory or offensive terminology, depiction of marginalized groups or outdated normative assumptions. However, the dataset itself does not contain personal identifiers, financial, health, biometric or government identification data.

**Any other comments?**

No further comments.

| Collection Process |
| --- |

**How was the data associated with each instance acquired?** Was the data directly observable (e.g., raw text, movie ratings), reported by subjects (e.g., survey responses), or indirectly inferred/derived from other data (e.g., part-of-speech tags, model-based guesses for age or language)? If data was reported by subjects or indirectly inferred/derived from other data, was the data validated/verified? If so, please describe how.

The data was directly acquired from Project Gutenberg, an online library of copyright-free e-books from the past few centuries. It also allows easy access by mirroring their entire catalogue or downloading their e-book collection via an API. The data was directly observable as raw texts.

**What mechanisms or procedures were used to collect the data (e.g., hardware apparatus or sensor, manual human curation, software program, software API)?** How were these mechanisms or procedures validated?

We have followed the official recommendation from Project Gutenberg to download the RDF files of the books via mirroring. The official mirror links can be found at their official website `https://www.gutenberg.org/help/mirroring.html`. We have also used their official repository `https://github.com/gutenbergtools` to interact with their resources when needed.

**If the dataset is a sample from a larger set, what was the sampling strategy (e.g., deterministic, probabilistic with specific sampling probabilities)?**

We have specifically focused on curating English texts from the period 1750–2000. However, in some cases, the original publication date of a work was either missing or inaccurately recorded. To address this, we employed an additional sampling strategy to ensure that works were properly curated and accurately categorized into their respective time intervals.

**Who was involved in the data collection process (e.g., students, crowdworkers, contractors) and how were they compensated (e.g., how much were crowdworkers paid)?**

Only the authors and co-authors were responsible for the collection of the data.

**Over what time-frame was the data collected? Does this time-frame match the creation time-frame of the data associated with the instances (e.g., recent crawl of old news articles)?** If not, please describe the time-frame in which the data associated with the instances was created.

The data was curated between October 2024 and March 2025. Since the dataset consists solely of historically published literary works, its content remains unaffected by the timeline of collection and compilation.

**Were any ethical review processes conducted (e.g., by an institutional review board)?** If so, please provide a description of these review processes, including the outcomes, as well as a link or other access point to any supporting documentation.

No

**Does the dataset relate to people?** If not, you may skip the remaining questions in this section.

Given its historical context, the data can relate to people or groups of individuals. The literary texts constituting Chronoberg represent biographies, works on important historical events between 1750 and 2000, social life, and similar aspects pertaining to historical society.

**Did you collect the data from the individuals in question directly, or obtain it via third parties or other sources (e.g., websites)?**

No, the data was not collected directly from any individual; rather, it was acquired from Project Gutenberg, a historical corpus of literary works, some of which may pertain to certain individuals.

**Were the individuals in question notified about the data collection?** If so, please describe (or show with screenshots or other information) how notice was provided, and provide a link or other access point to, or otherwise reproduce, the exact language of the notification itself.

No.

**Did the individuals in question consent to the collection and use of their data?** If so, please describe (or show with screenshots or other information) how consent was requested and provided, and provide a link or other access point to, or otherwise reproduce, the exact language to which the individuals consented.

The literary works obtained from Project Gutenberg are copyright-free and freely distributable. We have ensured that no information in these works was altered, preserving their integrity and originality. Given the historical time-frame covered by Chronoberg, the individuals referenced in these texts are not available to provide consent.

**If consent was obtained, were the consenting individuals provided with a mechanism to revoke their consent in the future or for certain uses?** If so, please provide a description, as well as a link or other access point to the mechanism (if appropriate).

Not applicable

**Has an analysis of the potential impact of the dataset and its use on data subjects (e.g., a data protection impact analysis) been conducted?** If so, please provide a description of this analysis, including the outcomes, as well as a link or other access point to any supporting documentation.

No such analysis with respect to data protection or privacy could be conducted as a) historical figures have long been deceased, b) information on historical figures has been disseminated in various historical works throughout time, c) Project Gutenberg has already provided a curated public archive that has excluded copy-righted and non-consensual material outside the public domain. Chronoberg derives itself from Project Gutenberg and thus not induce any new impact regarding data subjects.

**Any other comments?**

No further comments.

---

**Preprocessing/cleaning/labelling**

---

**Was any preprocessing/cleaning/labelling of the data done (e.g., discretization or bucketing, tokenization, part-of-speech tagging, SIFT feature extraction, removal of instances, processing of missing values)?** If so, please provide a description. If not, you may skip the remainder of the questions in this section.

We have followed several steps of preprocessing and labelling of our curated raw texts from Project Gutenberg.

- **Determination of original publication date:** We observed that e-books in Project Gutenberg often lack accurate publication years, or in some cases, the information is entirely missing. This step is particularly important for Chronoberg, as incorrect publication years would lead to misclassification of texts into the wrong temporal intervals.
- **Data partitioning:** The texts are grouped by year, resulting in 250 separate splits corresponding to individual years. In addition, we created broader bins spanning 50-year intervals.
- **Data filtering:** We removed all non-alphanumeric characters from the texts to facilitate easier adaptation across various downstream applications.

**Was the "raw" data saved in addition to the preprocessed/cleaned/labelled data (e.g., to support unanticipated future uses)?** If so, please provide a link or other access point to the "raw" data.

Yes, as a direct part of the dataset, we have made available two versions of Chronoberg: one consisting of the raw texts grouped by publication year, and another with processed texts, split into annotated sentences and likewise organized by publication year.

**Is the software used to preprocess/clean/label the instances available?** If so, please provide a link or other access point.

Yes, we will provide a link to a public GitHub explaining the preprocessing, cleaning, and labelling process. For reviewing purposes the code is attached as supplementary. Simultaneously, a Hugging Face link will also be provided for the dataset.

**Any other comments?**

No further comments.

| Uses |
| --- |

**Has the dataset been used for any tasks already?** If so, please provide a description.

The dataset has not been publicly available before. We highlight several potential downstream applications of Chronoberg in our accompanying experimental analysis, including: (i) continually adapting LLMs to historically evolving concepts, (ii) inspecting words and sentences within Chronoberg that have undergone diachronic shifts and outline future prospects, such as (iii) unlearning or modifying specific connotations in words used in contemporary English texts.

**Is there a repository that links to any or all papers or systems that use the dataset?** If so, please provide a link or other access point.

No.

**What (other) tasks could the dataset be used for?**

As part of our metadata, we also provide valence, arousal, and dominance scores for each sentence in Chronoberg, aiming to capture the affective sentiment expressed in the text. However, given the sensitivity of certain sentences, we strongly caution against interpreting these scores as definitive labels of positivity or negativity.

**Is there anything about the composition of the dataset or the way it was collected and preprocessed/cleaned/labelled that might impact future uses?** For example, is there anything that a future user might need to know to avoid uses that could result in unfair treatment of individuals or groups (e.g., stereotyping, quality of service issues) or other undesirable harms (e.g., financial harms, legal risks) If so, please provide a description. Is there anything a future user could do to mitigate these undesirable harms?

Alongside their publication year, we have also annotated the sentences in Chronoberg based on their VAD scores, denoting their affective polarity. However, we acknowledge that not every negatively

scored sentence based on the VAD score can be used as a harmful sentence. So, we encourage users to not treat the dataset for benchmarking hateful vs non-hateful applications. Similarly, as this data may directly or indirectly involve individuals, we discourage its use to specifically single out any individuals solely based on the VAD lexicons and affective polarity scores of the texts.

**Are there tasks for which the dataset should not be used?** If so, please provide a description.

Please refer to the previous question.

**Any other comments?**

No further comments.

| Distribution |
|:---:|

**Will the dataset be distributed to third parties outside of the entity (e.g., company, institution, organization) on behalf of which the dataset was created?** If so, please provide a description.

Yes, the dataset will be publicly available at HuggingFace, and the source code available at GitHub following de-anonymization post review.

**How will the dataset will be distributed (e.g., tarball on website, API, GitHub)** Does the dataset have a digital object identifier (DOI)?

We will make the dataset publicly available at HuggingFace.

**When will the dataset be distributed?**

The dataset will be distributed after de-anonymization.

**Will the dataset be distributed under a copyright or other intellectual property (IP) license, and/or under applicable terms of use (ToU)?** If so, please describe this license and/or ToU, and provide a link or other access point to, or otherwise reproduce, any relevant licensing terms or ToU, as well as any fees associated with these restrictions.

The dataset is distributed under the BSD 2-Clause "Simplified" License. It also comes under the full list of Project Gutenberg licenses, which can be found in `https://www.gutenberg.org/policy/license.html`

**Have any third parties imposed IP-based or other restrictions on the data associated with the instances?** If so, please describe these restrictions, and provide a link or other access point to, or otherwise reproduce, any relevant licensing terms, as well as any fees associated with these restrictions.

No.

**Do any export controls or other regulatory restrictions apply to the dataset or to individual instances?** If so, please describe these restrictions, and provide a link or other access point to, or otherwise reproduce, any supporting documentation.

No.

**Any other comments?**

No further comments.

| Maintenance |
|:---:|

**Who will be supporting/hosting/maintaining the dataset?**

Will be revealed after de-anonymization.

**How can the owner/curator/manager of the dataset be contacted (e.g., email address)?**

Will be revealed after de-anonymization.

**Is there an erratum?** If so, please provide a link or other access point.

There currently exists no erratum.

**Will the dataset be updated (e.g., to correct labelling errors, add new instances, delete instances)?** If so, please describe how often, by whom, and how updates will be communicated to users (e.g., mailing list, GitHub)?

Currently, no immediate updates are envisioned. If there appears to be an urgent need to update the dataset, the authors will be responsible for uploading a new version. We will update the version at HuggingFace and communicate the news on our website and HuggingFace.

**If the dataset relates to people, are there applicable limits on the retention of the data associated with the instances (e.g., were individuals in question told that their data would be retained for a fixed period of time and then deleted)?** If so, please describe these limits and explain how they will be enforced.

As the acquired data is copyright-free and free to distribute, we don't believe there is any applicable limits on the retention of the data associated with the instances.

**Will older versions of the dataset continue to be supported/hosted/maintained?** If so, please describe how. If not, please describe how its obsolescence will be communicated to users.

Yes, we will continue to host and support older versions of the dataset. HuggingFace, as a platform, supports versioning.

**If others want to extend/augment/build on/contribute to the dataset, is there a mechanism for them to do so?** If so, please provide a description. Will these contributions be validated/verified? If so, please describe how. If not, why not? Is there a process for communicating/distributing these contributions to other users? If so, please provide a description.

The code for data generation is publicly available on GitHub at [link provided after de-anonymization]. Validation/Verification of future contributions will not be in the scope of the authors. Yes, there are numerous possibilities to extend the dataset, such as:

- Extending beyond the 1750–2000 time-frame, the corpus can also be viewed as an ever-growing temporal dataset of historical contexts, given copy-right laws being based on 20 and 100 year cut-offs respectively.
- The dataset can be extended with several other languages besides Late Modern English.

**Any other comments?**

No further comments.