# OpenReview forum: "Chronoberg: Capturing Language Evolution And Temporal Awareness In Foundation Models"
_ICLR.cc/2026/Conference — Submitted to ICLR 2026_

### Official Review · Reviewer_m7ZD · 2025-10-20

**Soundness:** 2
**Presentation:** 3
**Contribution:** 2
**Rating:** 2
**Confidence:** 5

**Summary:**

This paper presents a dataset for evaluating temporal changes of words, which is then used to evaluate LLMs. Temporal change detection is a task that has been researched extensively (see SemEval workshops on this topic over the years for example) but its progress has been hindered by the lack of large-scale time-annotated datasets. This paper fill this important gap.

**Strengths:**

- The creation and release of a large-scale dataset for evaluating temporal changes of VAD of words is the main strength/contribution of this paper.
- The paper then goes on to evaluate multiple foundation models on this dataset.

**Weaknesses:**

- given the automatic nature of the annotation process, a manual verification (e.g. using a random sample) should be conducted.
- books are a curated set of texts and might not necessarily express the sentiment held by the broader population. Social media on the other hand would have been a better source for capturing modern VAD of words but cannot be used for historical texts. What is the coverage in terms of the number of authors covered in the corpus? If this number is low then it might not be representing the view point of the general public but an elite and small group of authors. This dataset bias should be investigated further.
- I am not sure whether all books in this corpus is suitable for this purpose of evaluating diachronic changes. For example, there could be fantasy books which do not reflect a social viewpoint but an imaginary one. Even if a book is written at a particular point in time, it might be handling a historical context that belongs to a different time period (e.g. a book written in 2020 on Edo period of Japan would not be reflecting the modern usage of Japanese language). I am not sure how these complications are handled in this corpus (or the authors are aware of such issues)?
- Although I appreciate the extensive evaluations conducted in the paper using the CHRONOBERG dataset, those findings will only be valid to the extent of the accuracy of the dataset itself.
- Moreover, I think this paper is more appropriate for the linguistic resources track at an NLP venue (e.g. LREC or xACL) rather than ICLR.

**Questions:**

- Did you conduct any manual (even at a small scale) evaluation of the VAD scores computed using Eq. (2)?
- What is the coverage in terms of the number of authors covered in the corpus? If this number is low then it might not be representing the view point of the general public but an elite and small group of authors. This dataset bias should be investigated further.
- In Table 1, can you explain the shift from neg to pos for `infatuation` and `destinty` please?
- What would the valency shift look like for a word such as `gay`, which is known to be used positively (happiness) in olden times, whereas more neutral? in modern usage?

---

> ### Author Response · Authors · 2025-11-20
>
> We thank Reviewer m7ZD for their insightful suggestions and review. Below, we address the concerns and questions raised.
>
> **[W1+Q1]** Manual verification due to automatic nature of annotation.
>
> Manual annotation of hate vs. non-hate is neither meaningful nor feasible in our setting. Hate-speech judgments are inherently subjective and highly context-dependent, an issue that becomes even more pronounced when spanning 250 years of historical texts. However, to address the concern, we have now included a quantitative experiment, shown in the table below, analysing how replacing words (e.g., “f\*ggot” with an innocuous synonym such as  “firewood” from Word2Vec) changes the classifier outputs.
> | Sentences                                                                                     | RoBERTa + Perspective API | OpenAI Moderation API | Valence Score | Affective Sentiment |
> |-----------------------------------------------------------------------------------------------|-----------------------|--------------------|---------------|--------------------|
> | In my way home to my tent, I saw a f\*ggot lying in the way.                                      | ❌                    | ❌                 | 0.05          | 👍                 |
> | In my way home to my tent, I saw a ~~f\*ggot~~ **firewood** lying in the way.                     | ✅                    | ✅                 | 0.10          | 👍                 |
> | In my way home to my tent, I saw a ~~f\*ggot~~ **sticks** lying in the way.                        | ✅                    | ✅                 | 0.02          | 👍                 |
> | In my way home to my tent, I saw a ~~f\*ggot~~ **bundle** lying in the way.                       | ✅                    | ✅                 | 0.09          | 👍                 |
> | In my way home to my tent, I saw a ~~f\*ggot~~ **turves** lying in the way.                       | ✅                    | ✅                 | 0.08          | 👍                 |
>
> **Note:** Here, ❌  denotes "hate", ✅ denotes "non-hate" and 👍 denotes positive sentiment. The results demonstrate that replacing the word "f\*ggot" with neutral alternatives changes the classification scores, highlighting the sensitivity of SOTA hatecheck models to surface-level keywords.
>
> We now sample 20 instances from each time period where the hate-speech classifiers and the VAD lexicon disagree, yielding a total of 100 sentences (initial 100% disagreement rate). We then apply the approach described above to these sentences and measure the resulting reduction in disagreement. The table below reports the updated disagreement rates between the VAD scores and the classifiers. Notably, this strategy substantially improves classifier performance, reducing disagreement to approximately $\sim 25$% for the OpenAI Moderation API and about $\sim 60$% for the Roberta + Perspective API.
>
> ### **Disagreement between VAD lexicons and hate-speech classifiers**
>
> | **Years**  | **OpenAI Moderation API**   | **Roberta**  **+ Perspective API**
> |-------------|-------------------|------------|
> | 1750–1799   | 25% | 65%               |
> | 1800–1849   | 30% | 70%               |
> | 1850–1899   | 25% | 70%               |
> | 1900–1950   | 25% | 65%               |
> | 1950–2000   | 15% | 50%               |
>
>
>
>
>
> **[W2+W3+Q2]** Books are a curated set of texts and might not reflect broader population sentiment. Author and genre coverage.
>
> We agree that books are curated and represent a subset of opinions, however, for historical periods, they remain the most comprehensive historical source available. This is not the case for social media, which is both modern and highly platform-specific. To address the concern about representativeness, we have added author coverage statistics, in addition to the full genre distribution and thematic composition, of which $\sim 24$% is fiction, with the rest being non-fiction (social life and customs, history, geo-politics, travel, etc). We have already reported a breakdown of categories in Appendix A.2 (please see Table 5). We note here that the category assignments here are derived from the Library of Congress subject headings and Project Gutenberg “bookshelf” tags. Additionally, we will include these annotations in the dataset.
>
> The most represented writers reflect the major publication trends of their respective eras. Out of the 20k authors in Chronoberg, overall, Mark Twain, Edward Bulwer-Lytton, George Manville Fenn, and George Alfred Henty are among the top contributors in Chronoberg, each with 80 - 120 works. We will add details regarding author coverage in the revised version. Regarding concerns about narrative settings that do not align with the publication period, we clarify that all analyses are anchored to the publication year, reflecting the linguistic usage of the time period, regardless of the book’s narrative setting.

---

> > ### Comment · Reviewer_m7ZD · 2025-11-22
> > **Reviewer response**
> >
> > Thank you for providing a detailed and clear response to my concerns/questions.
> >
> > - ** Manual Verification**:
> > *Manual annotation of hate vs. non-hate is neither meaningful nor feasible in our setting. Hate-speech judgments are inherently subjective and highly context-dependent...*
> > I disagree. I find it difficult to understand why something that you claim to be difficult to manually verify because it is "inherently subjective" but OK to do so when you use LLMs. How do you justify an automatic evaluation of something that is even subjective for humans to evaluate (according to your statement). The way manual evaluations are done by writing careful  annotation guidelines, getting the annotators to discuss their annotations and trying to improve the agreements. I know it is a hard thing but it is not anyway "infeasible". Hate speech is only one part of the issue and there are many other factors that result in for temporal changes of words.
> >
> > I am overly dissatisfied by the authors' unwillingness to accept obvious limitations regarding the use of books as a source for this analysis. There are much diverse historical documents beyond books to conduct such an analysis.
> >
> > I will keep my ratings.

---

> ### Author Response · Authors · 2025-11-20
>
> **[W4]** Findings are only valid to the extent of the dataset’s accuracy.
>
> As noted in the paper, we manually validated our publication year inference and obtained a mean absolute error of $\pm3.05$ years. This informed our choice of coarse 50-year temporal bins, which were selected to mitigate any variance. We reiterate this to clarify that the variance in publication date inference is substantially smaller than the temporal granularity at which Chronoberg and the experiments operate, and does not effect the robustness of the conclusions.
>
> **[W5]** Appropriateness of the venue.
>
> We acknowledge this subjective perspective. However, the primary contribution of this work lies in analysing continual learning and CL methods under temporal drift, and demonstrating how existing methods can help mitigate some of the pitfalls while remaining insufficient in others. Chronoberg serves as the foundation for these results, but the core focus is on learning dynamics rather than dataset creation, which we believe is well-aligned with the scope of ICLR.
>
> **[Q3]** The shift from negative to positive for "infatuation" and "destiny"?
>
> For infatuation: the diachronic shift stems from being associated with irrationality in terms of being unable to think clearly about the object of their infatuation, whereas in contemporary usage, it is often not purely negative and can convey excitement, passion or euphoria.
>
> For destiny: we can reasonably argue that the meaning of "destiny" evolved from a neutral or even ominous force ("irresistible necessity" to a more positive sense, often used in a aspirational, motivational or heroic sense. For instance, in the 1750s *destiny* appeared in more negatively connotated contexts, such as “had she woven her own destiny in the speculation of her wishes?”, whereas by the 1950s it was used in a more optimistic tone, as in “You are completely the master of your destiny, the captain of your soul”. Similarly, infatuation appeared in more negative contexts such as “I know not what infatuation instantaneously seized me,” whereas by the 1950s, it was used in a more neutral tone, as in “As my infatuation increased her coolness did not lessen; curiosity alone seemed to move her”.
>
> In the revised paper, we will add a brief explanation for each word in Table 1 to clarify these shifts. An extended set of examples are already included in the Appendix, and we will also include them in the main text for clarity.
>
> **[Q4]** Valence shift for a word like "gay"?
>
> We have analyzed the valence scores for the word "gay" across different eras and observed a gradual shift, from a highly positive score of 0.70 in the 1750s towards a more neutral score of 0.30 after the 1950s. This pattern aligns with the trend highlighted by the reviewer, confirming that "gay" historically carried a positive valence, while in modern usage it has become largely neutral. We thank the reviewer for pointing this out, we will incorporate it into the main text.

---

> ### Author Response · Authors · 2025-11-22
>
> We thank the reviewer for the feedback. We actually agree with the concerns regarding verification of annotations, and believe there may have been a misunderstanding regarding the points raised. Therefore, we would like to rectify this aspect and ensure it doesn't arise again.
>
> To be clear, we are stating that it is subjective and inappropriate to rely exclusively on LLMs for annotation tasks. Our entire argument is that LLM predictions are highly inconsistent, as they disagree significantly with each other (which we quantify in the paper), and often rely on superficial, surface-level cues that overlook historical context (which we quantify through the additional experiments that we provided in the rebuttal). As a result, we have introduced temporally adjusted VAD lexicons to better quantify such scenarios, thereby detaching ourselves from relying solely on LLM-assisted annotations. Also, note that our temporal VAD lexicons are derived from the contemporary VAD Lexicon by Saif et al., which provides human-rated scores across four primary dimensions of word meaning.
>
> In fact, we believe the latter experiment aligns closely with the reviewer’s request for manual verification. It focuses on analyzing instances that we have manually verified and know the model has misclassified, without assuming an oracle-like role for the entire dataset, since many cases are highly nuanced. For example, we manually verified the sentence “An outlandish pair bought from New York for some tory hussy,” which LLM classifiers incorrectly labeled as **non-hateful**, overlooking the historical meaning of hussy. However, replacing hussy with synonyms commonly used in modern times, following the same approach used for f*ggot, enabled the classifiers to identify it correctly.
>
> We hope this clarifies what we believe is a misunderstanding. If the reviewer has any specific suggestions on how we could modify the paper to prevent this in the future, we would be happy to make the necessary revisions. We agree with the reviewer’s concern and have been very careful in the paper to emphasise that automatic LLM annotations/predictions should never be relied upon exclusively.
>
> Lastly, we would like to reiterate that the section on quantifying hate flagging disagreements represents just a small portion of our work. We kindly ask you to also consider our large-scale, continual adaptation experiments, which are central to the paper's contributions.

---

### Official Review · Reviewer_bh7T · 2025-10-29

**Soundness:** 2
**Presentation:** 2
**Contribution:** 1
**Rating:** 4
**Confidence:** 3

**Summary:**

This paper presents Chronoberg, a large diachronic English corpus (1750–2000, 2.7B tokens) with sentence-level affective annotations and temporally aligned Valence–Arousal–Dominance (VAD) lexicons. The authors detail a pipeline for dating and cleaning Project Gutenberg texts, generate time-specific VAD lexicons via aligned embeddings, and use the resource to test hate-speech detectors and temporal adaptation of LLMs (sequential fine-tuning, EWC, LoRA). Results show severe forgetting and limited handling of historically shifting word valence, highlighting Chronoberg’s potential as a benchmark for temporal robustness and continual learning.

**Strengths:**

- Language temporal drift and historical semantics are important for ensuring LLM robustness.
- The paper presents interesting insights on semantic shift.

**Weaknesses:**

- Narrow domain coverage. Chronoberg is built entirely from Project Gutenberg books, spanning 1750–2000. This makes it a valuable literary resource but also limits generalization to modern or conversational language where current LLMs are deployed .

- Unclear LLM evaluation setup. The paper compares several classifiers, including “OpenAI” models, in its hate-speech evaluation tables . However, it does not describe how those models were prompted or whether they were informed about the historical origin of the text. Without that context, it’s hard to interpret what the observed disagreements actually reflect.

- Limited quantitative evaluation of temporal drift. Section 4.1 and the associated tables mostly illustrate qualitative examples of valence shifts and classifier disagreements. There is no formal metric (e.g., correlation or agreement score) quantifying alignment between the temporal VAD lexicons and model outputs .

- Unsurprising claims without measurement. The authors hypothesize that modern LLMs “rely too heavily on surface-level keywords,” but this remains an intuitive explanation rather than an empirically tested one . Quantifying how much this reliance contributes to misclassification would make the finding stronger.

- Vague notion of “contextualization.” The paper claims that temporally fine-tuned models “struggle with contextualization of historical content,” yet does not specify how contextual information (e.g., time metadata or retrieved examples) was provided during inference .

- Ambiguous terminology. Phrases like “dissonance of ~85% between OpenAI and RoBERTa” are reported without a clear definition of what “dissonance” measures (e.g., disagreement rate or correlation gap) . This makes quantitative interpretation difficult.

**Questions:**

- How do you expect findings from literary English (Chronoberg) to generalize to modern or conversational domains where current LLMs are applied?

- For the “OpenAI” model evaluations, what exact prompts or instructions were used? Was the model told the historical period or context of the text?

- Have you computed any quantitative metrics (e.g., correlation or agreement) between temporal VAD lexicons and classifier outputs, beyond qualitative examples?

- Can you operationalize “contextualization of historical content” — does it mean awareness of time, lexical shifts, or broader discourse cues?

- What precisely does the reported “∼85% dissonance” between OpenAI and RoBERTa measure? Is it disagreement rate, accuracy gap, or another metric?

---

> ### Author Response · Authors · 2025-11-20
>
> We would like to thank Reviewer bh7T for their thorough review and comments, which we address below.
>
> **[W1]** Narrow domain coverage, limiting generalization to modern conversational LLMs.
>
> We would like to respectfully disagree, since our goal with Chronoberg is not to tune LLMs to contemporary usage, an area already extensively explored in existing works, but rather to assess how models handle long-horizon continual learning in the presence of evolving semantic drift. The temporal structure of Chronoberg provides precisely the controlled setting required to study these dynamics, which is the central focus of our work.
>
> **[W2+Q2]** Unclear LLM evaluation setup.
>
> We clarify that "OpenAI" refers to the OpenAI Moderation API (https://platform.openai.com/docs/guides/moderation), a robust state-of-the-art hate speech detection API. Note that the API does not allow providing any additional context, such as the historical year of the text, during inference.
>
> Therefore, models were not provided with the historical year of the text as context during inference. However, this also reflects real-world scenarios where temporal metadata is often unavailable. We will state this in the revision.
>
> **[W3+Q3]** Limited quantitative evaluation of temporal drift. Lacking formal metrics quantifying alignment between VAD lexicons and model outputs.
>
> We have also conducted a quantitative assessment of the disagreement between lexical scores and the hate-speech classifier outputs, measuring the percentage of conflicting predictions. Notably, we found that these disagreements decrease as the language in our corpus transitions from the 1750s toward modern English connotations. In our revised paper, we will discuss this metric in more detail.
>
> We now sample 20 instances from each time period where the hate-speech classifiers and the VAD lexicon disagree, yielding a total of 100 sentences (initial 100% disagreement rate). We then apply the approach described above to these sentences and measure the resulting reduction in disagreement. The table below reports the updated disagreement rates between the VAD scores and the classifiers. Notably, this strategy substantially improves classifier performance, reducing disagreement to approximately $\sim 25$% for the OpenAI Moderation API and about $\sim 60$% for the Roberta + Perspective API.
>
> ### **Disagreement between VAD lexicons and hate-speech classifiers**
>    **Years**      | **OpenAI Moderation API** | **Roberta + Perspective API** |
>  |:--------------:|:--------------------------:|:-----------------------------:|
>  | 1750–1799      |            25%             |               65%             |
>  | 1800–1849      |            30%             |               70%             |
>  | 1850–1899      |            25%             |               70%             |
>  | 1900–1950      |            25%             |               65%             |
>  | 1950–2000      |            15%             |               50%             |
>
> In addition, as the continual learning experiments under temporal drift form a major part of the paper, we already provide quantitative evaluations in the form of confusion matrices, which show model performance across time intervals.

---

> > ### Author Response · Authors · 2025-11-21
> >
> > **[W4]** Unsurprising claims without measurement. No empirical quantification for the claim "LLMs rely too heavily on surface-level keywords".
> >
> > We would like to disagree that our findings are unsurprising. Our CL experiments demonstrate that models trained on Chronoberg struggle disproportionately with valence-shifting words, a phenomenon that has not been previously documented. This finding shows that models are sensitive to surface-level lexical cues whose meanings drift over time and that CL methods designed so far do not handle this scenario well, as they seem to assume that knowledge can strictly be accumulated.
> >
> > However, we agree that quantifying the contribution of keyword reliance would strengthen the result, and below, we have included an additional experiment in which we replace a problematic word with its benign synonym chosen via Word2Vec for a small set of sentences. The degree to which model predictions change under such substitutions would provide a direct quantitative measure of keyword dependence. For instance, the sentence “In my way home to my tent, I saw a f\*ggot lying in the way is unanimously flagged as hate by the hate-check classifiers. However, when we replaced the word “f\*ggot” with “firewood" (the actual meaning of the word in the 1800s) and other synonyms, the models no longer flagged the sentence.
> >
> > | Sentences                                                                                     | RoBERTa + Perspective API | OpenAI Moderation API | Valence Score | Affective Sentiment |
> > |-----------------------------------------------------------------------------------------------|-----------------------|--------------------|---------------|--------------------|
> > | In my way home to my tent, I saw a f\*ggot lying in the way.                                      | ❌                    | ❌                 | 0.05          | 👍                 |
> > | In my way home to my tent, I saw a ~~f\*ggot~~ **firewood** lying in the way.                     | ✅                    | ✅                 | 0.10          | 👍                 |
> > | In my way home to my tent, I saw a ~~f\*ggot~~ **sticks** lying in the way.                        | ✅                    | ✅                 | 0.02          | 👍                 |
> > | In my way home to my tent, I saw a ~~f\*ggot~~ **bundle** lying in the way.                       | ✅                    | ✅                 | 0.09          | 👍                 |
> > | In my way home to my tent, I saw a ~~f\*ggot~~ **turves** lying in the way.                       | ✅                    | ✅                 | 0.08          | 👍                 |
> >
> > **Note:** Here, ❌  denotes "hate", ✅ denotes "non-hate" and 👍 denotes positive sentiment. The results demonstrate that replacing the word "f\*ggot" with neutral alternatives changes the classification scores, highlighting the sensitivity of SOTA hatecheck models to surface-level keywords.
> >
> > **[W5+Q4]** Vague notion of contextualization / operationalize "contextualization of historical content".
> >
> > For better clarity, we will replace this phrasing with ”capture meaning at a specific point in time” in the revision. In the current experiments, models were not provided with any explicit temporal metadata. Our claim refers to the observation that even when fine-tuned on data divided into intervals, they often misinterpret words whose meanings have shifted across epochs. We will clarify this definition of contextualization in the paper, clarifying that no additional time metadata was provided.
> >
> > **[W6]** Ambiguous terminology : "$\sim 85$%" dissonance between OpenAI API and RoBERTa.
> >
> > The reported “dissonance” refers to the percentage of instances on which the
> > RoBERTa+Perspective API, OpenAI Moderation API, and VAD scores disagree in their predictions, allowing us to assess how modern classifiers handle historical language. We measure the same with respect to our VAD lexicons, since hate cannot be directly measured. We will clarify this in the revised manuscript.
> >
> > **[Q1]** Generalizing findings from literary English to modern English.
> >
> > Our focus is not on tuning LLMs for modern usage but on understanding how models adapt to semantic drift over time. Chronoberg’s controlled temporal structure allows us to isolate these dynamics, offering insights broadly applicable to any domain where language evolves.

---

> ### Comment · Reviewer_bh7T · 2025-11-24
>
> Thanks to the authors for the clarifications. Based on those, I decided to update my scores. Please make sure you include all the additional information in the final version.

---

### Official Review · Reviewer_Vpi9 · 2025-11-01

[review text omitted: it was posted to a different submission]

---

> ### Author Response · Authors · 2025-11-20
>
> We sincerely thank Reviewer Vpi9 for their detailed and constructive comments. Below we address their comments:
>
> **[W1]** Dependence on modern VAD lexicons introducing anachronisms.
>
> We agree that relying on a modern VAD lexicon may introduce anachronisms when inferring affective meaning in earlier time periods. In particular, we emphasize that our method uses modern valence scores as anchors to detect relative semantic drift through nearest-neighbour propagation in the embedding space of each 50-year time interval. We also caution against the interpretation of VAD values as absolute historical ground truth, but rather as indicators of diachronic shifts across epochs. In the revision, we are expanding the discussion to make this nuance explicit.
>
> **[W2]** Sequential training experiments only use Pythia-1.4b.
>
> We agree, and experiments with Pythia-160M are underway. The results are taking a while longer to finalise as they are computationally heavy, but they will be included in the revised version.
>
> **[Q1]** Contradiction between embedding-based VAD scores and established historical usage in literature.
>
> We agree that this is an important and interesting direction. However, as noted in the paper, this is very challenging. The reason being that there are few experts in this domain, and the scale of manual inspection required for a dataset of this size. We will be emphasizing this at the end of the paper and are actively exploring avenues to make this possible.
>
> **[Q2]** Transformer-based contextual embeddings instead of Word2Vec.
>
> We have not yet explored this, as Word2Vec suffices for our level of analysis. However, we recognize that this is an interesting future direction, and we will add it to our future work discussion.
>
> **[Q3]** Any hypotheses explaining why valence-shifting words degrade forward generalization more?
>
> We hypothesize that the stronger degradation in forward generalization for valence-shifting words is due to higher plasticity demands, since these words undergo substantial semantic change across epochs, requiring the model to update its internal representations when learning during later time intervals. This likely disrupts previously learned associations more than in the case of valence-stable words, whose meaning remain more or less consistent over time. We agree that this is an intriguing aspect of our findings and we will be adding this discussion to the conclusion.

---

### Official Review · Reviewer_GhzC · 2025-11-07

**Soundness:** 3
**Presentation:** 3
**Contribution:** 3
**Rating:** 6
**Confidence:** 3

**Summary:**

The paper addresses the crucial problem that most Large Language Models (LLMs) are trained on temporally stationary data, limiting their ability to comprehend the long-term evolution of language, social norms, and semantics (diachronic variation). The introduction of Chronoberg, a large-scale (2.7B tokens) and temporally structured corpus of full-length English book texts (25,061 books from Project Gutenberg) spanning 250 years (1750–2000). Experiments using the dataset show that LLMs trained sequentially struggle significantly with catastrophic forgetting and generalization to future language, particularly concerning words that have undergone affective valence shifts. This positions Chronoberg as a necessary benchmark for temporal generalization, Continual Learning, and evaluating the temporal robustness of AI systems.

**Strengths:**

1. The introduction of Chronoberg is a major contribution, filling a critical gap in LLM training and evaluation. It provides a large-scale (2.7B tokens) and long-horizon (250 years) temporally structured corpus of full-length texts.
2. The systematic construction of temporally calibrated VAD lexicons for $\sim$335,000 words is highly innovative.

**Weaknesses:**

1. The temporal structure, which is the foundation of the dataset, relies on an inferred publication date from external sources (OpenLibrary), while validated, this process has an unavoidable Mean Absolute Error (MAE) of $\pm 3.05$ years.

2. The methodology for constructing the temporal VAD lexicons relies on selecting the Top-K nearest neighbors ($K=20$). However, the main text does not include a systematic ablation study demonstrating how the choice of $K$ (or the effect of the 50-year interval size) impacts the final, crucial results, such as the LLM perplexity gap between valence-stable and valence-shifting test sets.

**Questions:**

1. How did you get the "positive" or "negative" in Table 1? It is still not clear to me.

2. The paper uses Word2Vec and CADE alignment. Could the authors justify why this combination was chosen over more contemporary diachronic methods?

---

> ### Author Response · Authors · 2025-11-20
>
> We sincerely thank Reviewer GhzC for their thoughtful and constructive feedback. Below we address their comments:
>
> **[W1]** Publication year estimate introduces MAE of $\pm$3.05 years.
>
> We agree with the reviewer. We explicitly highlighted this limitation and caution users to choose the granularity carefully, recommending coarse intervals of at least 10-15 years to ensure robustness. Still, we would like to clarify that while the inferred publication years introduce an MAE of $\pm 3.05 $ years, this level of noise is negligible at the temporal granularity at which Chronoberg is designed to be used. We will discuss this more prominently in the revised paper.
>
> **[W2]** No systematic ablation for top-K neighbourhood size and 50-year interval choice.
>
> We thank the reviewer for raising this point. We would like to point the reviewer to Appendix A.3, which provides a discussion on the choice of $K$, with Fig. 7 and 8 visualizing results for $K \in {5,20, 100, 500}$. We observe that smaller neighbourhoods (eg., $K=5$) produce sharper, localized shifts in valence scores, while larger neighbourhoods (eg., $K=500$) naturally average out the signal and substantially reduce observable variation. As a result, large $K$ values would also diminish observable differences in the continual learning (CL) experiments, while smaller $K$ values would reflect sharper shifts.
>
> Further, we agree that a smaller temporal resolution is informative, and we will add an additional 20-year interval ablation for lexicon construction and corresponding analysis in the revised version. While we would also like to include corresponding CL experiments at this finer granularity, doing so would require training a large number of models, and is not feasible. We emphasize, however, that the added lexicon analysis will further demonstrate the robustness of our findings.
>
> **[Q1]** “Positive” or “negative” labels in Table 1.
>
> These labels are derived from the valence values produced by our VAD scoring (valence $> 0 = $ positive, valence $< 0 =$ negative. We would like to emphasize here that we follow this sign convention for simplicity when illustrating semantic shifts, and we are aware that the connotation of a word can depend on context and subjective interpretation. Our goal is not to assert categorial labels but to highlight how VAD scoring captures affective shifts that are interpretable over time.
>
> **[Q2]** Choice of Word2Vec + CADE over newer diachronic models.
>
> We chose Word2Vec + CADE because the combination is a well-validated, computationally lightweight baseline that reliably scales to 250 years of data and allows for quantitative measures of long-term diachronic analysis. This setup is also compatible with our nearest-neighbour valence propagation method, which relies on stable alignment across time. We will clarify these motivations more explicitly in the revised paper.

---

### Author Response · Authors · 2025-12-02
**Urgent request: one of the reviews seems to have been rolled back in a problematic way (to a review of a different paper)**

Dear PCs, SAC, and AC,

we would like to humbly request for a (potentially manual) correction of one of our reviews, where a mistake in rolling back seems to have happened.

Specifically, reviewer Vpi9 seems to have initially posted a different paper's review for our work. Clearly, this was an accident and the reviewer had noticed it and updated the review promptly to the correct one for our paper (for which we thank the reviewer). This was before any discussions had happened or rebuttals had been posted.

However, it seems the review was now rolled back to the very first thing the reviewer posted, i.e. not the review for our paper. This is unfortunate, because the real review was factual and gave an 8/10 score.

We are very concerned that this rollback to the state of a review that doesn't belong to our paper is going to very negatively impact our paper, because the logical conclusion of any AC would now be to discard it entirely in their consideration. We believe this would be very unfair and would never have happened, had the hard rollback not occurred.

We kindly request the PCs to manually exchange the review to the correct one.
If this is not possible, we would humbly request to be allowed to withdraw the paper without it continuing to appear on open review, as clearly too much of the review process would be compromised.
However, as ICLR is dear to us, we would strongly prefer the solution that corrects the mistake again.

---

### Author Response · Authors · 2025-12-03
**Summary of Manuscript Revisions**

We would like to sincerely thank the reviewers for their feedback and comments. We have addressed all concerns with the following updates and improvements, and submit an improved manuscript, with changes highlighted in blue. In particular, we have included:

-   Page 3: Discussion of the effect of 3.05 MAE more prominently.

-   Page 4: Choice of Word2Vec + CADE over newer diachronic models.

-   Page 4: Discussion on modern VAD lexicons introducing anachronisms.

-   Page 6: Added explanation for diachronic shifts for words in Table 1. The appendix (page 21) now has the VAD scores for the \"gay\" example.

-   Page 7: LLM evaluation setup with respect to context.

-   Page 7 (Table 2b) and Page 8 (Table 3): Disagreement between VAD and hate-speech classifiers and Quantitative results showing reliance of modern LLMs on surface-level keywords.

-   Updated \"OpenAI\" to \"OpenAI Moderation API\".

-   Page 8: Ambiguous wording in \"contextualization of historical content\" replaced with \"capturing meaning at a specific point in time\".

-   Page 8: Definition of \"disagreement\" metric added.

-   Page 10: Explicitly state that stronger degradation in forward generalization for valence shifting words is due to plasticity demands.

-   Page 20, Table 6 and 7 (Appendix) :Added Genre distribution and Author representation in Chronoberg.

-   Page 22, Table 10 (Appendix): Added Ablation with 20 year interval  for VAD lexicons.

-   Page 26, Figure 11 (Page 28): Added Pythia-160m experiments.

We believe our responses have adequately addressed all major concerns raised by reviewers, as reflected in the improved and maintained positive scores. Thank you again for the time invested into reviewing our work.

---

### Author Response · Authors · 2025-12-03
**Overview of the rebuttal before the leakage**

Dear PCs, SAC, and AC,

We recognize the present situation and appreciate the course of action suggested by ICLR. We sincerely appreciate the time and effort dedicated to the coordination of this discussion phase. To assist in the evaluation of our work, we provide a summary of the rebuttals submitted prior to the leak for your reference:
-   **Reviewer Vpi9 (score of 8)**: The review initially did not correspond to our submission. However, this was corrected shortly after (on November 13) and the correct one can be found in the review revision history. We kindly request that you assess the corrected review along with our response.\
    As requested in **\[W1\]**, we added a discussion on modern VAD lexicons introducing anachronisms. For **\[W2&Q3\]**, **we added experiments with Pythia 160M**, discussed how higher plasticity demands lead to degradation of forward generalization for valence-shifting words. We clarified the choice of Word2Vec+CADE for **\[Q2\]** and addressed **\[Q1\]** on VAD scores and historical usage.

-   **Reviewer bh7T (initial score of 4, updated to 6)**: For **\[W2+Q2\]** regarding LLM evaluation setup, we clarified in our response that
    \"OpenAI\" refers to \"OpenAI Moderation API\", and there is no option to provide historical year as context. Regarding **\[W3+Q3\]** on quantitative evaluation, we added 2 new experiments (along with existing CL experiments) showing how surface-level keyword replacements (e.g., \"f\*ggot\" → \"firewood\") reduced classifier disagreement. For **\[W6+Q1\]**, we have clarified ambiguous terminology in this work as well as our focus on the study of how models adapt to semantic drift over time.\
    The reviewer confirmed satisfaction, **increasing their score to 6**, confirming this in a comment. We mention this to ensure that the updated score is kindly taken into account in the final assessment.

-   **Reviewer GhzC (score of 6)**: We addressed the reviewers concerns in **\[W1\]** regarding the MAE of 3.05 years due to publication
    year estimation, clarifying that this does not effect results at the temporal granularity of our experiments. For **\[W2\]**, we clarified that ablation for top-K neighbourhood size are already provided in the appendix, and additionally report **changes in valence scores derived from lexicons built using Word2Vec models trained on 20-year temporal slices.** in the revision. We have added clarification regarding labels in Table 1 addressing **\[Q1\]**. For **\[Q2\]**, we clarify in the revised manuscript that the choice of Word2Vec+CADE is due to its computational efficiency, scalability to 250 years and compatibility with our nearest-neighbour valence propagation method requiring stable alignment.

-   **Reviewer m7ZD (score of 2)**: In response to **\[W1&Q1\]**, we clarified that manual annotation of hate speech is not feasible due to its subjective and context-dependent nature across 250 years of historical texts. We included a quantitative experiment showing how replacing surface-level words (e.g., "f\*ggot" with "firewood") affects classifier outputs and decreases disagreement between hate-speech classifiers and VAD scores. For **\[W2&W3&Q2\]** regarding the use of books as a data source, we explained that books are the most comprehensive historical data source. We have **added author and genre statistics** in the response and revision. Regarding dataset accuracy in **\[W4\]**, we clarified that the MAE of $\pm3.05$ years is negligible relative to the 50-year temporal bins and does not affect the robustness of our results. For **\[Q3&Q4\]**, we explained the sentiment shifts for words as requested and included the valence shift for \"gay\" in the revision. **In response to the follow-up comment**, we clarified the misunderstanding about manual verification of annotations. We agree with the reviewer that relying solely on **LLMs is inappropriate due to their inconsistency** and failure to account for historical context, as emphasized in the paper. We use temporally adjusted VAD lexicons(based on human-rated scores) to avoid relying solely on LLM  annotations. To this end, we provide experiments where **LLMs misclassified nuanced instances** and how **substituting modern synonyms allowed classifiers to classify them correctly**. To reiterate, in addition to the hate-flagging disagreement experiments, our paper also includes large-scale continual adaptation experiments, which are central to the contributions.

We truly appreciate the careful consideration of these changes, we have posted a detailed summary of changes in the revision in the comment below. It is unfortunate that the reviewers do not have the opportunity to respond directly to the changes, as we have made the exact updates they requested. Thank you again for your attention to these clarifications. We are grateful for your effort in these challenging circumstances, and look forward to the next steps in the review process.

---

### Meta-Review · Area_Chair_w5eg · 2026-01-05

**Summary:**

The paper introduces Chronoberg, a large-scale (2.7B tokens), temporally structured corpus curated from Project Gutenberg, spanning the years 1750–2000 . The authors enrich this corpus with temporally calibrated Valence-Arousal-Dominance (VAD) lexicons constructed via diachronic Word2Vec models to track semantic drift . Through extensive experiments using foundation models (Pythia architecture), the paper investigates two main phenomena: (1) the failure of modern hate-speech detectors (like OpenAI Moderation API) due to anachronistic interpretations of historical text , and (2) the difficulty of standard Continual Learning (CL) methods (like EWC and LoRA) in handling "valence-shifting" words, which leads to catastrophic forgetting and poor forward generalization .

**Rebuttal Conclusion**:
 Despite the authors' responsive rebuttal and the inclusion of additional experiments (e.g., Pythia-160M scaling analysis, synonym substitution tests), the final decision is Reject. While the dataset fills a gap in long-horizon temporal resources, the fundamental methodology for establishing "ground truth" lacks the strictness required for a standard benchmark. The main obstacle, as emphasized by Reviewer m7ZD, is the complete reliance on automated tagging (modern VAD lexicons + embedding alignment) without expert human verification. Consequently, the paper is viewed as a resource contribution with validity risks, better suited for a specialized NLP venue (e.g., LREC or ACL Resources) where dataset curation standards can be more strictly evaluated, rather than a representation learning conference like ICLR.

**Reviewer Concerns:**

**Concerns Addressed by the Rebuttal**
•**Model Scale and Generalization**: Addressed by providing additional experiments with Pythia-160M in the appendix, demonstrating that the observed trends in catastrophic forgetting and plasticity demands hold across different model scales. This satisfied Reviewer Vpi9's concern regarding the exclusive use of Pythia-1.4B.
•**Keyword Reliance and Interpretability**: Resolved by introducing a synonym substitution experiment (e.g., replacing "faggot" with "firewood" in historical contexts) . This quantitatively demonstrated that modern classifiers rely on surface-level keywords, addressing Reviewer bh7T’s request for real proof supporting the "keyword reliance" hypothesis.
•**Evaluation Metrics and API Setup**: Addressed by clarifying that the OpenAI Moderation API does not support context injection (years), which simulates real-world "blind" moderation scenarios . The authors also added a specific "Disagreement" metric table to quantify the divergence between VAD scores and classifiers, satisfying Reviewer bh7T.

**Outstanding Concerns**:
•**Lack of Ground Truth Verification**: This remains the primary ground for rejection. Reviewer m7ZD maintained a strong objection that the dataset relies entirely on unsupervised algorithms (Word2Vec) to generate historical sentiment scores without any manual validation by domain experts (historians/linguists) . In the context of a benchmark paper, the inability to verify whether the generated "semantic drifts" are historically accurate or artifacts of the embedding process is a critical flaw.
•**Methodological Anachronism**: As noted by Reviewer Vpi9 and acknowledged by the authors, using modern NRC VAD lexicons as anchors to score 18th-century text introduces systematic anachronisms . While the authors argue this serves as a relative metric, it fundamentally undermines the reliability of the "ground truth" used to evaluate model performance. A benchmark cannot robustly evaluate "temporal awareness" if its own labels are biased by modern usage.
•**Venue Fit and Contribution Type**: Reviewer m7ZD argued that the paper is primarily a dataset release (Resource Paper) rather than a methodological advance in representation learning. Since the machine learning contributions are limited to applying standard baselines (EWC, LoRA) to this new data, the work lacks the algorithmic novelty typically expected at ICLR.

**Reviewer Scores:**

The final opinions are split: 8, 6, 6, 2.
 Reviewer Vpi9 (Score: 8) strongly supported the work for its novelty in addressing temporal drift. Reviewers GhzC and bh7T (Score: 6) were convinced by the rebuttal experiments regarding data noise and classifier reliance, raising their assessment to marginally above acceptance. However, Reviewer m7ZD (Score: 2) maintained a strong rejection, arguing that the "infeasibility" of manual verification is not a valid excuse for releasing an unverified benchmark in a top-tier conference. The AC ultimately sides with the rigorous stance on data validity.

---

### Decision · Program_Chairs · 2026-01-26

Reject